# Counterfactual Learning of Stochastic Policies with Continuous Actions

**Houssam Zenati**[*]                                                  *h.zenati@ucl.ac.uk*
*Gatsby Computational Neuroscience Unit, University College London*

**Alberto Bietti**                                            *abietti@flatironinstitute.org*
*Center for Computational Mathematics, Flatiron Institute, New York, NY, USA*

**Matthieu Martin**                                               *mat.martin@criteo.com*
**Eustache Diemert**                                                *e.diemert@criteo.com*
*Criteo AI Lab*

**Pierre Gaillard**                                             *pierre.gaillard@inria.fr*
**Julien Mairal**                                                *julien.mairal@inria.fr*
*Univ. Grenoble Alpes, Inria, CNRS, Grenoble INP, LJK, 38000 Grenoble, France*

**Reviewed on OpenReview:** *https://openreview.net/forum?id=fC4bh1PmZr*

## Abstract

Counterfactual reasoning from logged data has become increasingly important for many applications such as web advertising or healthcare. In this paper, we address the problem of learning stochastic policies with continuous actions from the viewpoint of counterfactual risk minimization (CRM). While the CRM framework is appealing and well studied for discrete actions, the continuous action case raises new challenges about modelization, optimization, and offline model selection with real data which turns out to be particularly challenging. Our paper contributes to these three aspects of the CRM estimation pipeline. First, we introduce a modelling strategy based on a joint kernel embedding of contexts and actions, which overcomes the shortcomings of previous discretization approaches. Second, we empirically show that the optimization aspect of counterfactual learning is important, and we demonstrate the benefits of proximal point algorithms and smooth estimators. Finally, we propose an evaluation protocol for offline policies in real-world logged systems, which is challenging since policies cannot be replayed on test data, and we release a new large-scale dataset along with multiple synthetic, yet realistic, evaluation setups.

## 1 Introduction

Logged interaction data is widely available in many applications such as drug dosage prescription (Kallus & Zhou, 2018), recommender systems (Li et al., 2012), or online auctions (Bottou et al., 2013). An important task is to leverage past data in order to find a good *policy* for selecting actions (*e.g.*, drug doses) from available features (or *contexts*), rather than relying on randomized trials or sequential exploration, which may be costly to obtain or subject to ethical concerns.

More precisely, we consider offline logged bandit feedback data, consisting of contexts and actions selected by a given *logging policy*, associated to observed rewards. This is known as *bandit feedback*, since the reward is only observed for the action chosen by the logging policy. The problem of finding a good policy thus requires a form of *counterfactual* reasoning to estimate what the rewards would have been, had we used a different policy. When the logging policy is stochastic, one may obtain unbiased reward estimates under a

---

[*]Most of this work was done while the author was at the Criteo AI Lab and Inria.

new policy through importance sampling with inverse propensity scoring (IPS, Horvitz & Thompson, 1952). One may then use this estimator or its variants for optimizing new policies without the need for costly experiments (Bottou et al., 2013; Dudik et al., 2011; Swaminathan & Joachims, 2015a;b), an approach also known as counterfactual risk minimization (CRM). While this setting is not sequential, we assume that learning a *stochastic* policy is required so that one may gather new exploration data after deployment.

In this paper, we focus on stochastic policies with continuous actions, which, unlike the discrete setting, have received little attention in the context of counterfactual policy optimization (Demirer et al., 2019; Kallus & Zhou, 2018; Chen et al., 2016). As noted by Kallus & Zhou (2018) and as our experiments confirm, addressing the continuous case with naive discretization strategies performs poorly. Our first contribution is about *data modeling*: we introduce a joint embedding of actions and contexts relying on kernel methods, which takes into account the continuous nature of actions, leading to rich classes of estimators that prove to be effective in practice.

In the context of CRM, the problem of *estimation* is intrinsically related to the problem of *optimization* of a non-convex objective function. In our second contribution, we underline the role of optimization algorithms (Bottou et al., 2013; Swaminathan & Joachims, 2015b). We believe that this aspect was overlooked, as previous work has mostly studied the effectiveness of estimation methods regardless of the optimization procedure. In this paper, we show that appropriate tools can bring significant benefits. To that effect, we introduce differentiable estimators based on soft-clipping the importance weights, which are more amenable to gradient-based optimization than previous hard clipping procedures (Bottou et al., 2013; Wang et al., 2017). We provide a statistical analysis of our estimator and discuss its theoretical performance with regards to the literature. We also find that proximal point algorithms (Rockafellar, 1976) tend to dominate simpler off-the-shelf optimization approaches, while keeping a reasonable computation cost.

Finally, an open problem in counterfactual reasoning is the difficult question of reliable *evaluation* of new policies based on logged data only. Despite significant progress thanks to various IPS estimators, we believe that this issue is still acute, since we need to be able to estimate the quality of policies and possibly select among different candidate ones *before* being able to deploy them in practice. Our last contribution is a small step towards solving this challenge, and consists of a new offline evaluation benchmark along with a new large-scale dataset, which we call CoCoA, obtained from a real-world system. The key idea is to introduce importance sampling diagnostics (Owen, 2013) to discard unreliable solutions along with significance tests to assess improvements to a reference policy. We believe that this contribution will be useful for the research community as we are unaware of similar publicly available large-scale datasets for continuous actions.

Our findings, outlined below, are based on experimental results and validated through diverse tasks, demonstrating the effectiveness of counterfactual learning for stochastic policies with continuous actions.

1. *Discrete methods are not effective enough for continuous action problems.* Discretization strategies tend to underperform in applications involving continuous actions. To address this, we propose a modeling approach that jointly embeds contexts and continuous actions, demonstrating superior empirical performance compared to the linear models of Kallus & Zhou (2018); Chen et al. (2016) and the tree-based method of (Majzoubi et al., 2020). This approach, termed the Counterfactual Loss Predictor (CLP), is detailed in Section 3.2.

2. *Optimization perspectives of counterfactual learning problems matter.* Our work introduces a novel differentiable estimator based on soft-clipping of importance weights. Empirically, the soft-clipping strategy outperforms existing weight transformation methods (Wang et al., 2017; Su et al., 2019; Metelli et al., 2021). Additionally, our findings underscore the effectiveness of proximal point methods (Rockafellar, 1976), which have been shown to be particularly beneficial for optimizing nonconvex objective functions (Paquette et al., 2018). The soft-clipping estimator and the associated optimization tools are introduced in Section 4. In Section 5, we analyze the excess risk of the proposed estimator and compare it to existing methods in the literature.

3. *Real-world counterfactual risk minimization requires a standard offline evaluation protocol.* To address this, we release CoCoA, a large-scale dataset designed for counterfactual learning with continuous actions, and propose an offline evaluation procedure for model selection in counterfactual

risk minimization tasks. Our protocol is empirically validated for assessing and identifying unreliable solutions, making it a crucial tool for real-world deployment of policies learned through counterfactual risk minimization. The dataset and evaluation protocol are presented in Section 6.

## 2 Related Work

A large effort has been devoted to designing CRM estimators that have less variance than the IPS method, through clipping importance weights (Bottou et al., 2013; Wang et al., 2017), variance regularization (Swaminathan & Joachims, 2015a), or by leveraging reward estimators through doubly robust methods (Dudik et al., 2011; Robins & Rotnitzky, 1995). In order to tackle an overfitting phenomenon termed "propensity overfitting", Swaminathan & Joachims (2015b) also consider self-normalized estimators (Owen, 2013). Such estimation techniques also appear in the context of sequential learning in contextual bandits (Agarwal et al., 2014; Langford & Zhang, 2008), as well as for off-policy evaluation in reinforcement learning (Jiang & Li, 2016). In contrast, the setting we consider is not sequential. Moreover, unlike direct approaches (Dudik et al., 2011) which learn a cost predictor to derive a deterministic greedy policy, our approach learns a model indirectly by rather minimizing the policy risk. In discrete actions settings, several approaches have been proposed for large action spaces (Majzoubi et al., 2020; Sachdeva et al., 2023; Saito et al., 2023).

While most approaches for counterfactual policy optimization tend to focus on discrete actions, few works have tackled the continuous action case, again with a focus on estimation rather than optimization. In particular, propensity scores for continuous actions were considered by Hirano & Imbens (2004). More recently, evaluation and optimization of continuous action policies were studied in a non-parametric context by Kallus & Zhou (2018), Lee et al. (2022), and by Demirer et al. (2019) in a semi-parametric setting. In causal inference, recent works have addressed the question of treatment effect estimation with continuous actions (Colangelo & Lee, 2023; Bockel-Rickermann et al., 2023).

In contrast to these previous methods, (i) we focus on stochastic policies while they consider deterministic ones, even though the kernel smoothing approach of Kallus & Zhou (2018) and Lee et al. (2022) may be interpreted as learning a deterministic policy perturbed by Gaussian noise. (ii) The terminology of *kernels* used by Kallus & Zhou (2018) refers to a different mathematical tool than the kernel embedding used in our work. We use positive definite kernels to define a nonlinear representation of actions and contexts in order to model the reward function, whereas Kallus & Zhou (2018) use *kernel density estimation* to obtain good importance sampling estimates and not model the reward. Chen et al. (2016) also use a kernel embedding of contexts in their policy parametrization, while our method jointly models contexts and actions. Moreover, their method requires computing an $n \times n$ Gram matrix, which does not scale with large datasets; in principle, it should be however possible to modify their method to handle kernel approximations such as the Nyström method (Williams & Seeger, 2001). Besides, their learning formulation with a quadratic problem is not compatible with CRM regularizers introduced by (Swaminathan & Joachims, 2015a;b) which would change their optimization procedure. Eventually, we note that Krause & Ong (2011) use similar kernels to ours for jointly modeling contexts and actions, but in the different setting of sequential decision making with upper confidence bound strategies. (iii) While Kallus & Zhou (2018) and Demirer et al. (2019) focus on policy *estimation*, our work introduces a new continuous-action data *representation* and encompasses *optimization*: in particular, we propose a new contextual policy parameterization, which leads to significant gains compared to baselines parametrized policies on the problems we consider, as well as further improvements related to the optimization strategy. We also note that, apart from Demirer et al. (2019) that uses an internal offline cross-validation for model selection, previous works did not perform offline model selection nor evaluation protocols, which are crucial for deploying methods on real data.

Optimization methods for learning stochastic policies have been mainly studied in the context of reinforcement learning through the policy gradient theorem (Ahmed et al., 2019; Sutton et al., 2000; Williams, 1992). Such methods typically need to observe samples from the new policy at each optimization step, which is not possible in our setting. Other methods leverage a form of off-policy estimates during optimization (Kakade & Langford, 2002; Schulman et al., 2017), but these approaches still require to deploy learned policies at each step, while we consider objective functions involving only a fixed dataset of collected data. In the context of CRM, Su et al. (2019) introduce an estimator with a continuous clipping objective that achieves an improved

bias-variance trade-off over the doubly-robust strategy. Nevertheless, this estimator is non-smooth, unlike our soft-clipping estimator.

## 3 Modeling of Continous Action Policies

We now review the CRM framework, originally introduced for discrete action spaces, and then present our modeling approach for policies with continuous actions.

### 3.1 Background

For a stochastic policy $\pi$ over a set of actions $\mathcal{A}$, a contextual bandit environment generates i.i.d. context features $x \sim \mathcal{P}_{\mathcal{X}}$ in $\mathcal{X}$, actions $a \sim \pi(\cdot|x)$ and losses $y \sim \mathcal{P}_{\mathcal{Y}}(\cdot|x, a)$, which may be seen as negative rewards, for some conditional probability distribution $\mathcal{P}_{\mathcal{Y}}$. We denote the resulting distribution over triplets $(x, a, y)$ by $\mathcal{P}_\pi$. Then, the risk of a policy $\pi$ is defined as:

$$L(\pi) = \mathbb{E}_{(x,a,y)\sim\mathcal{P}_\pi}\left[y\right]. \tag{1}$$

For the logged bandit problem, the environment provides an offline *logged* dataset $(x_i, a_i, y_i)_{i=1,\dots,n}$, where we assume $(x_i, a_i, y_i) \sim \mathcal{P}_{\pi_0}$ i.i.d. for a given stochastic *logging*[1] policy $\pi_0$, and we assume the *propensities* $\pi_{0,i} := \pi_0(a_i|x_i)$ to be known.[2] Then, the task lies in using the logged data to determine a policy $\hat{\pi}$ in a set of *stochastic* policies $\Pi$ with small risk.[3] Unlike reinforcement learning problems, where data can be gathered iteratively, here the data is collected at once using $\pi_0$. As a result, the quality of a new policy $\hat{\pi}$ can only be assessed counterfactually: "How would $\hat{\pi}$ have performed if it had been used during data collection?"

To minimize $L$, we typically have access to an empirical estimator $\hat{L}$ allowing to perform counterfactual reasoning and solve the following regularized problem:

$$\hat{\pi} \in \underset{\pi\in\Pi}{\arg\min}\left\{\hat{L}(\pi) + \Omega(\pi)\right\}, \tag{2}$$

where $\Omega$ is a regularizer. With appropriate counterfactual estimators for $\hat{L}$, the framework of (2) has been called *counterfactual risk minimization* (Swaminathan & Joachims, 2015a).

In this paper, we are motivated by real-world problems involving continuous actions, such as those presented in Section 7, which requires addressing three aspects:

- *Modeling:* Which policy class $\Pi$ is suitable for continuous actions, particularly when it is important to capture potential nonlinear interactions between actions and contexts? To address this, we propose the Counterfactual Loss Predictor (CLP), a method that serves as a smooth approximation of a greedy policy, leveraging a kernel embedding of actions and contexts.

- *Optimization*: We propose a counterfactual loss functions $\hat{L}$ that is differentiable (Section 4.1). Then, we discuss optimization strategies to deal with the non-convexity of $\hat{L}$ and $\Omega$ (Section 4.2).

- *Evaluation:* In Section 6, design a new offline protocol to evaluate the performance of the solution of (2) and to perform model selection on the choice $(\Pi, \hat{L}, \Omega)$.

Before, we present our contributions, we recall below existing choices for $\hat{L}$ and $\Omega$ that we will evaluate later, and another classical estimator $\hat{\pi}$ (Direct method) which will motivate our new policy class $\Pi$.

---

[1]The terminology associated to "logged" data and "logging" policy from Swaminathan & Joachims (2015a) is often referred to as trajectory data and behavior policy in the importance sampling literature (Owen, 2013)

[2]In the continuous action case, we assume that the probability distributions admit densities and thus propensities denote the density function of $\pi_0$ evaluated on the actions given a context.

[3]Note that this definition may also include deterministic policies by allowing Dirac measures, unless $\Pi$ includes a specific constraint *e.g.*, minimum variance, which may be desirable in order to gather data for future offline experiments.

**Counterfactual policy learning.** The counterfactual[4] approach tackles the distribution mismatch between the logging policy $\pi_0(\cdot|x)$ and a policy $\pi$ in $\Pi$ via importance sampling. The IPS method (Horvitz & Thompson, 1952) relies on correcting the distribution mismatch using the well-known relation

$$L(\pi) = \mathbb{E}_{(x,a,y)\sim\mathcal{P}_{\pi_0}}\left[y\frac{\pi(a|x)}{\pi_0(a|x)}\right], \tag{3}$$

assuming $\pi_0$ has non-zero mass on the support of $\pi$. This allows us to derive an unbiased empirical estimate

$$\hat{L}_{\text{IPS}}(\pi) = \frac{1}{n}\sum_{i=1}^{n} y_i \frac{\pi(a_i|x_i)}{\pi_{0,i}}. \tag{4}$$

However, the empirical estimator $\hat{L}_{\text{IPS}}(\pi)$ has large variance and is subject to various overfitting phenomena. First, this estimator can be prone to *loss overfitting* which refers to overweighting losses $y_i$ of rare sample points which were highly unlikely under $\pi_0$. For this, regularization strategies have been proposed, such as clipping the importance sampling weights (cIPS, Bottou et al., 2013; Swaminathan & Joachims, 2015a). Second, such estimators also suffer from *propensity overfitting* (Swaminathan & Joachims, 2015b) where the minimization of the latter estimator can lead to learn policies that artificially overfit training data for negative losses $y_i$ (to maximize $\pi(a_i|x_i)$) or conversely avoid training data when losses are positives. Subsequently, *self-normalized* estimators (SNIPS, Swaminathan & Joachims, 2015b; Owen, 2013) have been presented to circumvent this issue, see Appendix A for a review and Appendix B.1 for a motivating example.

**The direct method DM and the optimal greedy policy.** For this class of methods, an important quantity is the expected loss given actions and context, denoted by $\eta^*(x,a) = \mathbb{E}[y|x,a]$. If this expected loss was known, an optimal (deterministic) greedy policy $\pi^*$ would simply select actions that minimize the expected loss

$$\pi^*(x) = \arg\min_{a\in\mathcal{A}} \eta^*(x,a). \tag{5}$$

It is then tempting to use the available data to learn an estimator $\hat{\eta}(x,a)$ of the expected loss, for instance by using ridge regression to fit $y_i \approx \hat{\eta}(x_i,a_i)$ on the training data. Then, we may use the deterministic greedy policy

$$\hat{\pi}_{\text{DM}}(x) = \arg\min_{a\in\mathcal{A}} \hat{\eta}(x,a).$$

This approach, termed *direct method* (DM), has the benefit of avoiding the high-variance problems of IPS-based methods, but may suffer from large bias since it ignores the potential mismatch between $\hat{\pi}_{\text{DM}}$ and $\pi_0$. Specifically, the bias is problematic when the logging policy provides unbalanced data samples (*e.g.*, only samples actions in a specific part of the action space) leading to overfitting (Bottou et al., 2013; Dudik et al., 2011; Swaminathan & Joachims, 2015b). Conversely, counterfactual methods re-balance these generated data samples with importance weights and mitigate the distribution mismatch to better estimate reward function on less explored actions (see explanations in Appendix C). Nevertheless, such loss estimators can be sometimes effective in practice and may be used to improve IPS estimators in the so-called doubly robust (DR) estimator (Dudik et al., 2011) by applying IPS to the residuals $y_i - \hat{\eta}(x_i,a_i)$, thus using $\hat{\eta}$ as a control variate to decrease the variance of IPS.

### 3.2 The Counterfactual Loss Predictor (CLP) for Continuous Actions Policies

We recall that our estimator $\hat{\pi}$ is designed by optimizing (2) over a class of policies $\Pi$. In this subsection, we discuss how to choose $\Pi$ to model continuous actions problems. We emphasize that when considering continuous action spaces, the choice of policies $\Pi$ is more involved than in the discrete case. One may indeed naively discretize the action space into buckets and leverage discrete action strategies, but then local information within each bucket gets lost and it is non-trivial to choose an appropriate bucketization of the action space based on logged data, which contains non-discrete actions.

---

[4]The "counterfactual" term comes from the causal inference literature and refers in this policy learning context to all estimators using the inverse propensity scoring (IPS) method (Horvitz & Thompson, 1952)

We focus on stochastic policies belonging to certain classes of continuous distributions, such as Normal or log-Normal with context-dependent means and variances. Specifically, we consider a set of policies of the form

$$\Pi = \left\{ \pi_\theta \quad \text{s.t. for any } x \in \mathcal{X}, \quad \pi_\theta(\cdot|x) = \mathcal{D}(\mu_\beta(x), \sigma^2) \quad \text{with} \quad \theta = (\beta, \sigma) \in \Theta \right\}, \tag{6}$$

where $\mathcal{D}(a, b)$ is a distribution of probability with mean $a$ and variance $b$, such as the Normal distribution, and $\Theta$ is a parameter space. Here, the parameter space $\Theta$ can be written as $\Theta = \Theta_\beta \times \Theta_\sigma$. The space $\Theta_\sigma$ is either a singleton (if $\sigma$ is considered as a fixed parameter specified by the user) or $\mathbb{R}_+^*$ (if $\sigma$ is a parameter to be optimized). The space $\Theta_\beta$ is the parameter space which models the contextual mean $x \mapsto \mu_\beta(x)$, which will be specified shortly. Note that in our policy set, the variance parameter $\sigma$ is a parameter that calibrates the stochasticity of the policy. For applications where a learned policy needs to be deployed to collect data in later rounds, a minimal variance can be enforced to ensure future exploration.

### 3.2.1 Examples of simple context-dependent policies

Before introducing a more expressive model for $\Theta_\beta$, we consider the following simple counterfactual baselines for $\Theta_\beta$ that only consider contexts and that will be compared to our model in the experimental Section 7. Given a context $x$ in $\mathcal{X} \subset \mathbb{R}^d$:

- *constant*: $\mu_\beta(x) = \beta$      (context-independent);
- *linear*: $\mu_\beta(x) = \langle x, \beta_1 \rangle + \beta_0$ with $\beta = (\beta_0, \beta_1) \in \mathbb{R}^{d+1}$;
- *poly*: $\mu_\beta(x) = \langle xx^\top, \beta_1 \rangle + \beta_0$ with $\beta = (\beta_0, \beta_1) \in \mathbb{R}^{d^2+1}$.

These baselines require learning the parameters $\beta$ by using the CRM approach (2). Intuitively, the goal is to find a stochastic policy that is close to the optimal deterministic one from Eq. (5). Yet, these approaches do not model potential interactions between actions and contexts. While these approaches, adopted by Chen et al. (2016), Kallus & Zhou (2018) can be effective in simple problems, they may be limited in more difficult scenarios where the expected loss $\eta^*(x, a)$ has a complex behavior as a function of $a$. To address this issue, we introduce the Counterfactual Loss Predictor (CLP) model in the next section.

### 3.2.2 The counterfactual loss predictor (CLP) model

The CLP model relies on two ideas: (i) setting $\mu_\beta(x)$ to be a smooth approximation of the greedy policy (5), and (ii) using a parametric model $\eta_\beta(x, a)$ of the loss which is expressive enough to capture complex interactions between actions and contexts. More precisely, assuming that we are given such a parametric model $\eta_\beta(x, a)$, which we call *loss predictor*, we parametrize the mean of a stochastic policy by using a soft-argmin operator with temperature $\gamma > 0$:

$$\text{CLP:} \qquad \mu_\beta^{\text{CLP}}(x) = \sum_{i=1}^m a_i \frac{\exp(-\gamma \eta_\beta(x, a_i))}{\sum_{j=1}^m \exp(-\gamma \eta_\beta(x, a_j))}, \tag{7}$$

where $a_1, \ldots, a_m \in \mathcal{A}$ are anchor points (*e.g.*, a regular grid or quantiles of the action space). Then, $\mu_\beta$ may be viewed as a smooth approximation of the greedy policy $\mu_{\text{greedy}}(x) = \arg\min_a \eta(x, a)$. This allows CLP policies to capture complex behavior of the expected loss as a function of $a$. The motivation for introducing a soft-argmin operator over a finite number of anchor points is to build a policy that avoids the explicit optimization over the continuous action set, while making the resulting CRM problem differentiable.

Next, we need of course to define the loss predictor $\eta_\beta(x, a)$. We choose it of the form

$$\eta_\beta(x, a) = \langle \beta, \psi(x, a) \rangle,$$

for some parameter $\beta \in \mathbb{R}^p$, which norm controls the smoothness of $\eta_\beta$, and a feature map $\psi(x, a) \in \mathbb{R}^p$ that we detail in two parts: a joint kernel embedding between the actions and the contexts and a Nyström approximation. The complete modeling of $\eta_\beta(x, a)$ is summarized in Figure 1.

*1. Joint kernel embedding.* In a continuous action problem, a reasonable assumption is that losses vary smoothly as a function of actions. Thus, a good choice is to take $\eta$ in a space of smooth functions such as the

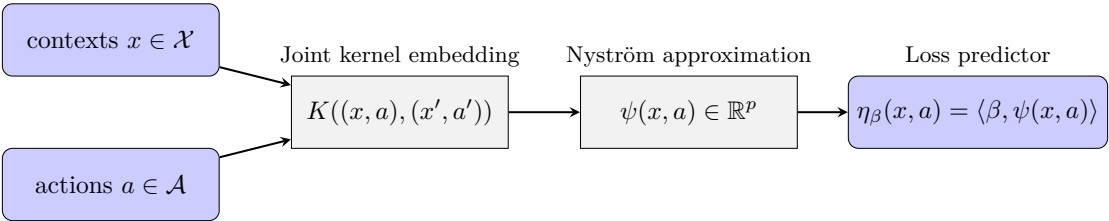

Figure 1: Illustration of the joint kernel embedding for the counterfactual loss predictor (CLP) and loss estimator.

reproducing kernel Hilbert space $\mathcal{H}$ (RKHS) defined by a positive definite kernel (Schölkopf & Smola, 2002), so that one may control the smoothness of $\eta$ through regularization with the RKHS norm. More precisely, we consider kernels of the form

$$K((x,a),(x',a')) = \langle \psi_{\mathcal{X}}(x), \psi_{\mathcal{X}}(x') \rangle e^{-\frac{\alpha}{2}\|a-a'\|^2}, \tag{8}$$

where, for simplicity, $\psi_{\mathcal{X}}(x)$ is either a linear embedding $\psi_{\mathcal{X}}(x) = x$ or a quadratic one $\psi_{\mathcal{X}}(x) = (xx^T, x)$, while actions are compared via a Gaussian kernel, allowing to model complex interactions between contexts and actions.

*2. Nyström method and explicit embedding.* Since traditional kernel methods lack scalability, we rely on the classical Nyström approximation (Williams & Seeger, 2001) of the Gaussian kernel, which provides us a finite-dimensional approximate embedding $\psi_{\mathcal{A}}(a)$ in $\mathbb{R}^m$ such that $e^{-\frac{\alpha}{2}\|a-a'\|^2} \approx \langle \psi_{\mathcal{A}}(a), \psi_{\mathcal{A}}(a') \rangle$ for all actions $a, a'$. This allows us to build a finite-dimensional embedding

$$\psi(x,a) = \psi_{\mathcal{X}}(x) \otimes \psi_{\mathcal{A}}(a), \tag{9}$$

where $\otimes$ denotes the tensorial product, such that

$$K((x,a),(x',a')) \approx \langle \psi_{\mathcal{X}}(x), \psi_{\mathcal{X}}(x') \rangle \langle \psi_{\mathcal{A}}(a), \psi_{\mathcal{A}}(a') \rangle = \langle \psi(x,a), \psi(x',a') \rangle.$$

More precisely, Nyström's approximation consists in projecting each point from the RKHS to a $m$-dimensional subspace defined as the span of $m$ anchor points, representing here the mapping to the RKHS of $m$ actions $\bar{a}_1, \bar{a}_2, \ldots, \bar{a}_m$ of the Nyström dictionary $\mathcal{Z}$. In practice, we may choose $\bar{a}_i$ to be equal to the $a_i$ in (7), since in both cases the goal is to choose a set of "representative" actions. For one-dimensional actions ($\mathcal{A} \subseteq \mathbb{R}$), it is reasonable to consider a uniform grid, or a non-uniform ones based on quantiles of the empirical distribution of actions in the dataset. In higher dimensions, one may simply use a K-means algorithms and assign anchor points to centroids. The number of anchor points directly affects the quality of the approximation: a larger number enhances representational power but reduces computational efficiency for large-scale applications, and vice versa. In practice, this number is treated as a hyperparameter, selected using the offline protocol described in Section 6. Its impact is further illustrated through a comparison between continuous and discrete strategies in Appendix G.1.

From an implementation point of view, Nyström's approximation considers the embedding $\psi_{\mathcal{A}}(a) = K_{\mathcal{Z}\mathcal{Z}}^{-1/2} K_{\mathcal{Z}}(a)$, where $K_{\mathcal{Z}\mathcal{Z}} = [K_{\mathcal{A}}(\bar{a}_i, \bar{a}_j)]_{ij}$ and $K_{\mathcal{Z}}(a) = [K_{\mathcal{A}}(a, \bar{a}_i)]_i$ and $K_{\mathcal{A}}$ is the Gaussian kernel. The whole procedure to design the CLP policy set $\Pi_{\text{CLP}}$ is given in Algorithm 1.

The anchor points that we use can be seen as the parameters of an *interpolation* strategy defining a smooth function, similar to knots in spline interpolation. Naive discretization strategies would prevent us from exploiting such a smoothness assumption on the loss with respect to actions and from exploiting the structure of the action space. Note that Section 7 provides a comparison with naive discretization strategies, showing important benefits of the kernel approach. Our goal was to design a stochastic, computationally tractable, differentiable approximation of the optimal (but unknown) greedy policy (5).

---

**Algorithm 1:** Construction of the CLP Policy Set

---

**Input:** Temperature $\gamma > 0$, kernel $K$, Nyström dictionary $\mathcal{Z} \subseteq \mathcal{A}$, parametric distribution $\mathcal{D}$ (such as Normal or log-Normal).

**Output:** Policy set $\Pi_{\text{CLP}}$.

1. Define the $d$-dimensional feature map $\psi$ as in Eq. (9) by using $K$ and $\mathcal{Z}$.

2. For any $\beta \in \mathbb{R}^d$ and $(x, a) \in \mathcal{X} \times \mathcal{A}$, define

$$\eta_\beta(x, a) = \langle \beta, \psi(x, a) \rangle \qquad \text{and} \qquad \mu_\beta^{\text{CLP}}(x) = \sum_{a \in \mathcal{Z}} \frac{\exp(-\gamma \eta_\beta(x, a))}{\sum_{a' \in \mathcal{Z}} \exp(-\gamma \eta_\beta(x, a'))}.$$

3. Define the policy set

$$\Pi_{\text{CLP}} = \left\{ \pi \text{ s.t. } \forall x \in \mathcal{X}, \ \pi(\cdot|x) = \mathcal{D}(\mu_\beta^{\text{CLP}}(x), \sigma^2), \text{ with } (\beta, \sigma) \in \Theta \right\}.$$

---

## 4 On Optimization Perspectives for CRM

Because our models yield non-convex CRM problems, it is crucial to study optimization aspects. Here, we introduce a differentiable clipping strategy for importance weights and discuss optimization algorithms.

### 4.1 Soft Clipping IPS

The classical hard clipping estimator

$$\hat{L}_{\text{cIPS}}(\pi) = \frac{1}{n} \sum_{i=1}^{n} y_i \min \left\{ \pi(a_i|x_i)/\pi_{0,i}, M \right\} \tag{10}$$

makes the objective function non-differentiable, and yields terms in the objective with clipped weights to have zero gradient. In other words, a trivial stationary point of the objective function is that of a stochastic policy that differs enough from the logging policy such that all importance weights are clipped. To alleviate this issue, we propose a differentiable logarithmic soft-clipping strategy. Given a threshold parameter $M \geq 0$ and an importance weight $w_i = \pi(a_i|x_i)/\pi_{0,i}$, we consider the soft-clipped weights:

$$\zeta(w_i, M) = \begin{cases} w_i & \text{if } w_i \leq M \\ \alpha(M) \log (w_i + \alpha(M) - M) & \text{otherwise,} \end{cases} \tag{11}$$

where $\alpha(M)$ is such that $\alpha(M) \log(\alpha(M)) = M$, which yields a differentiable operator. We illustrate in Fig. 2 the expression of the logarithmic clipping

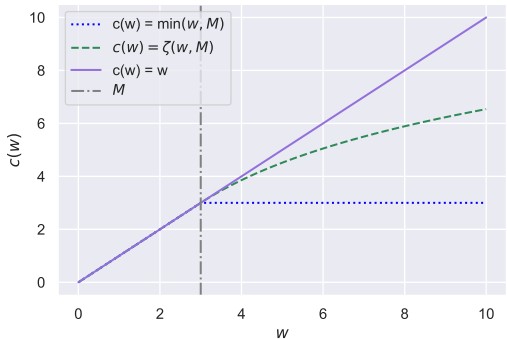

Figure 2: Soft Clipping of importance sampling weight $w$.

We give further explanations about the benefits of clipping strategies in Appendix B and compare our soft-clipping strategy to other methods in Appendix G.2. Then, the soft clipping IPS (scIPS) estimator is defined as:

$$\hat{L}_{\text{scIPS}}(\pi) = \frac{1}{n} \sum_{i=1}^{n} y_i \zeta \left( \frac{\pi(a_i|x_i)}{\pi_{0,i}}, M \right). \tag{12}$$

Note that our soft-clipping estimator differs from existing importance weight transformation strategies such as the power-mean correction of importance sampling (PMCIS) (Metelli et al., 2021), the SWITCH (Wang et al., 2017) and CAB (Su et al., 2019) estimators, which were not motivated by differentiability issuesas (see an empirical comparison in Appendix G.2).

We now provide a similar generalization bound to that of Swaminathan & Joachims (2015a) (for the hard-clipped version) for the variance-regularized objective of this soft-clipped estimator, justifying its use as a good optimization objective for minimizing the expected risk. Writing $\chi_i(\pi) = y_i \zeta \left( \frac{\pi(a_i|x_i)}{\pi_0(a_i|x_i)}, M \right)$, we recall the empirical variance with scIPS that is used for regularization:

$$\hat{V}_{\text{scIPS}}(\pi) = \frac{1}{n-1} \sum_{i=1}^{n} (\chi_i(\pi) - \bar{\chi}(\pi))^2, \qquad \text{with} \quad \bar{\chi}(\pi) = \frac{1}{n} \sum_{i=1}^{n} \chi_i(\pi). \tag{13}$$

We assume that costs $y_i \in [-1, 0]$ almost surely, as in (Swaminathan & Joachims, 2015a), and make the additional assumption that the importance weights $\pi(a_i|x_i)/\pi_0(a_i|x_i)$ are upper bounded by a constant $W$ almost surely for all $\pi \in \Pi$. This is satisfied, for instance, if all policies have a given compact support (*e.g.*, actions are constrained to belong to a given interval) and $\pi_0$ puts mass everywhere in this support.

**Proposition 4.1** (Generalization bound for $\hat{L}_{\text{scIPS}}(\pi)$). *Let $\Pi$ be a policy class and $\pi_0$ be a logging policy, under which an input-action-cost triple follows $\mathcal{D}_{\pi_0}$. Assume that $-1 \leq y \leq 0$ a.s. when $(x, a, y) \sim \mathcal{D}_{\pi_0}$ and that the importance weights are bounded by $W$. Then, with probability at least $1 - \delta$, the IPS estimator with soft clipping (12) on $n$ samples satisfies*

$$\forall \pi \in \Pi, \qquad L(\pi) \leq \hat{L}_{scIPS}(\pi) + O \left( \sqrt{\frac{\hat{V}_{scIPS}(\pi)\left(C_n(\Pi, M) + \log \frac{1}{\delta}\right)}{n}} + \frac{S\left(C_n(\Pi, M) + \log \frac{1}{\delta}\right)}{n} \right),$$

*where $S = \zeta(W, M) = O(\log W)$, $\hat{V}_{scIPS}(\pi)$ denotes the empirical variance of the cost estimates (20), and $C_n(\Pi, M)$ is a complexity measure of the policy class defined in (24).*

We prove Proposition 4.1 in Appendix E. This generalization error bound motivates the use of the empirical variance penalization as in (Swaminathan & Joachims, 2015a) and shows that minimizing both the empirical risk and penalization of the soft clipped estimator minimizes an upper bound of the true risk of the policy.

Note that while the bound requires importance weights bounded by a constant $W$, the bound only scales logarithmically with $W$ when $W \gg M$, compared to a linear dependence for IPS (Swaminathan & Joachims, 2015a). However we gain significant benefits in terms of optimization by having a smooth objective.

**Remark 4.1.** *If losses are in the range $[-c, 0]$, the constant $S$ can be replaced by $cS$, making the bound homogeneous in the scale (indeed, the variance term is also scaled by $c$).*

**Remark 4.2.** *For a fixed parameter $M$, the scIPS estimator is less biased than the cIPS. Indeed, we can bound the importance weights as $\min\left\{ \frac{\pi(a|x)}{\pi_0(a|x)}, M \right\} \leq \zeta\left( \frac{\pi(a|x)}{\pi_0(a|x)}, M \right) \leq \frac{\pi(a|x)}{\pi_0(a|x)}$ and subsequently derive the bound:*

$$\left| \mathbb{E}_{x, a \sim \pi_0(\cdot|x)} \left[ y \min\left\{ \frac{\pi(a|x)}{\pi_0(a|x)}, M \right\} - y \right] \right| \geq \left| \mathbb{E}_{x, a \sim \pi_0(\cdot|x)} \left[ y \zeta\left( \frac{\pi(a|x)}{\pi_0(a|x)}, M \right) - y \right] \right| \geq 0$$

*We note however that the parameter $M$ may have different optimal values for both methods, and that the main motivation for such a clipping strategy is to provide a differentiable estimator which is not the case for cIPS in areas where all point are clipped.*

## 4.2 Proximal Point Algorithms

Non-convex CRM objectives have been optimized with classical gradient-based methods (Swaminathan & Joachims, 2015a;b) such as L-BFGS (Liu & Nocedal, 1989), or the stochastic gradient descent approach (Joachims et al., 2018). Proximal point methods are classical approaches originally designed for convex optimization (Rockafellar, 1976), which were then found to be useful for nonconvex functions (Fukushima & Mine, 1981; Paquette et al., 2018). In order to minimize a function $\mathcal{L}$, the main idea is to approximately solve a sequence of subproblems that are better conditioned than $\mathcal{L}$, such that the sequence of iterates converges towards a better stationary point of $\mathcal{L}$. More precisely, for our class of parametric policies, the proximal point method consists of computing a sequence

$$\theta_k \approx \arg\min_\theta \left( \mathcal{L}(\pi_\theta) + \frac{\kappa}{2} \|\theta - \theta_{k-1}\|_2^2 \right), \tag{14}$$

where $\mathcal{L}(\pi_\theta) = \hat{L}(\pi_\theta) + \Omega(\pi_\theta)$ and $\kappa > 0$ is a constant parameter. The regularization term $\Omega$ often penalizes the variance (Swaminathan & Joachims, 2015b), see Appendix F.2. The role of the quadratic function in (14) is to make subproblems "less nonconvex" and for many machine learning formulations, it is even possible to obtain convex sub-problems with large enough $\kappa$ (see Paquette et al., 2018).

In this paper, we consider such a strategy (14) with a parameter $\kappa$, which we set to zero only for the last iteration. Note that the effect of the proximal point algorithm differs from the proximal policy optimization (PPO) strategy used in reinforcement learning (Schulman et al., 2017), even though both approaches are related. PPO encourages a new stochastic policy to be close to a previous one in Kullback-Leibler distance. Whereas the term used in PPO modifies the objective function (and changes the set of stationary points), the proximal point algorithm optimizes and finds a stationary point of the original objective $\mathcal{L}$, even with fixed $\kappa$.

The proximal point algorithm (PPA) introduces an additional computational cost as it leads to solving multiple sub-problems instead of a single learning problem. In practice for 10 PPA iterations and with the L-BFGS solver, the computational overhead was about $3\times$ in comparison to L-BFGS without PPA. This overhead seems to be the price to pay to improve the test reward and obtain better local optima, as we show in the experimental section 7.2.2. Nevertheless, we would like to emphasize that computational time is often not critical for the applications we consider, since optimization is performed offline.

## 5 Analysis of the Excess Risk

In the previous section, we have introduced a new counterfactual estimator $\hat{L}_{\text{scIPS}}$ (12) of the risk, which satisfies good optimization properties. Motivated by the generalization bound in Proposition 4.1, for any policy class $\Pi$, we associate $\hat{L}_{\text{scIPS}}$ with the data-dependent regularizer and define the following CRM estimator

$$\hat{\pi}^{CRM} = \arg\min_{\pi \in \Pi} \left\{ \hat{L}_{\text{scIPS}}(\pi) + \lambda \sqrt{\frac{\hat{V}_{\text{scIPS}}(\pi)}{n}} \right\}, \tag{15}$$

where $\hat{V}_{\text{scIPS}}(\pi)$ is the empirical variance defined in (13). In this section, we provide theoretical guarantees on the excess risk of $\hat{\pi}^{CRM}$, first for any general policy class $\Pi$, then for our newly introduced policy class $\Pi_{\text{CLP}}$ (Section 3.2). We provide the following high-probability upper-bound on the excess-risk.

**Proposition 5.1** (Excess risk upper bound). *Consider the notations and assumptions of Proposition 4.1. Let $\hat{\pi}^{CRM}$ be the solution of the CRM problem in Eq. (15) and $\pi^* \in \Pi$ be any policy. Then, with well chosen parameters $\lambda$ and $M$, denoting the variance $\sigma_*^2 = \text{Var}_{\pi_0}\big[\pi^*(a|x)/\pi_0(a|x)\big]$, with probability at least $1 - \delta$:*

$$L(\hat{\pi}^{CRM}) - L(\pi^*) \lesssim \sqrt{\frac{(1 + \sigma_*^2)\log(W + e)\big(C_n(\Pi, M) + \log\frac{1}{\delta}\big)}{n}} + \frac{\log(W + e)(C_n(\Pi, M) + \log\frac{1}{\delta})}{n},$$

*where $\lesssim$ hides universal multiplicative constants. In particular, assuming also that $\pi_0(x|a)^{-1}$ are uniformly bounded, the complexity of the class $\Pi_{CLP}$ described in Section 3.2, applied with a bounded kernel and $\Theta = \big\{\beta \in \mathbb{R}^m, \ s.t \ \|\beta\| \leq C\big\} \times \{\sigma\}$, is of order*

$$C_n(\Pi_{CLP}) \leq O(m \log n),$$

*where $m$ is the size of the Nyström dictionary and $O(\cdot)$ hides multiplicative constants independent of $n$ and $m$.*

The proof and the exact definition of $C_n(\Pi, M)$ are provided in Appendix E. Our analysis relies on Theorem 15 of Maurer & Pontil (2009).

**Comparison with related work**  The closest works are the ones of Chen et al. (2016) and Kallus & Zhou (2018). Chen et al. (2016) analyze their method for Besov policy classes $B_{1,\infty}^\alpha(\mathbb{R}^d)$. When $\alpha \to \infty$, they obtain a rate of order $\mathcal{O}(n^{-1/4})$. In this case, their setting is parametric and their rate can be compared to our $O(n^{-1/2})$ when $m$ is finite. Kallus & Zhou (2018) provide bounds with respect to general deterministic classes of functions, whose complexity is measured by their Rademacher complexity. For parametric classes, their excess risk is bounded (up to logs) by $R_{\hat\pi} \lesssim h^{-2}n^{-1/2} + h^{-1}n^{-1/2} + h^2$, where $h$ is a smoothing parameter. By optimizing the bandwidth $h = \mathcal{O}(n^{-1/8})$, their method also yields a rate of order $\mathcal{O}(n^{-1/4})$.

Yet, a key difference between their setting and ours explains the gap between their rate $\mathcal{O}(n^{-1/4})$ and $\mathcal{O}(n^{-1/2})$ of Proposition 5.1. Both consider deterministic policy classes, while we only consider stochastic policies. Indeed, $W$ and $\sigma^*$ would be unbounded for deterministic policies in Proposition 5.1. Therefore, to leverage deterministic policies, they both need to smooth their predictions and suffer an additional bias that we do not incur. This is why there is a difference between their rate and ours. For instance, for stochastic classes with variance $\sigma^2$, Kallus & Zhou (2018) would satisfy $R_{\hat\pi} \lesssim \sigma^{-2}n^{-1/2} + \sigma^{-1}n^{-1/2}$ for $h \approx \sigma$, which would also entail a rate of order $\mathcal{O}(n^{-1/2})$. Interestingly, on the other hand, our approach would satisfy a rate $O(n^{-1/3})$ for deterministic policies, i.e., $\sigma^2 \to 0$ (see Appendix E.2). This may be explained by the fact that, contrary to Kallus & Zhou (2018); Chen et al. (2016) who only use it in practice, we consider variance regularization and clipping in our analysis.

Another related work is (Demirer et al., 2019). They obtain an excess risk rate of $\mathcal{O}(n^{-1/2})$ when learning deterministic continuous action policies with a policy space of finite and small VC-dimension. Under a margin condition, as in bandit problems, their rate may be improved to $O(\log(n)/n)$. However, their method significantly differs from ours and Chen et al. (2016), Kallus & Zhou (2018) because it relies on a two steps plug-in procedure: first estimate a nuisance function, then learn a policy using with a value function using this estimate. Eventually, we note that Majzoubi et al. (2020) also enjoys a regret of $\mathcal{O}(n^{-1/2})$ (up to logarithmic factors) but learns tree policies that are hardly comparable to ours. Both approaches turn out to perform worse in all our benchmarks, as seen in Section. 7.2.2.

# 6  On Evaluation and Model Selection for Real-World Data

The CRM framework helps finding solutions when online experiments are costly, dangerous or raising ethical concerns. As such it needs a reliable validation and evaluation procedure before rolling-out any solution in the real world. In the continuous action domain, previous work have mainly considered semi-simulated scenarios (Bertsimas & McCord, 2018; Kallus & Zhou, 2018), where contexts are taken from supervised datasets but rewards are synthetically generated. To foster research on practical continuous policy optimization, we release a new large-scale dataset called CoCoA, which to our knowledge is the first to provide logged exploration data from a real-world system with continuous actions. Additionally, we introduce a benchmark protocol for reliably evaluating policies using off-policy evaluation.

## 6.1  The CoCoA Dataset

The CoCoA dataset comes from the Criteo online advertising platform which ran an experiment involving a randomized, continuous policy for real-time bidding. Data has been properly anonymized so as to not disclose any private information. Each sample represents a bidding opportunity for which a multi-dimensional context $x$ in $\mathbb{R}^d$ is observed and a continuous action $a$ in $\mathbb{R}^+$ has been chosen according to a stochastic policy $\pi_0$ that is logged along with the reward $-y$ (meaning cost $y$) in $\mathbb{R}$. The reward represents an advertising objective such as sales or visits and is jointly caused by the action and context $(a, x)$. Particular care has been taken to guarantee that each sample $(x_i, a_i, \pi_0(a_i|x_i), y_i)$ is independent. The goal is to learn a contextual, continuous, stochastic policy $\pi(a|x)$ that generates more reward in expectation than $\pi_0$, evaluated offline,

while keeping some exploration (stochastic part). As seen in Table 1, a typical feature of this dataset is the high variance of the cost ($\mathbb{V}[Y]$), motivating the scale of the dataset $N$ to obtain precise counterfactual estimates.

| $N$ | $d$ | $\mathbb{E}[-Y]$ | $\mathbb{V}[Y]$ | $\mathbb{V}[A]$ | $\mathbb{P}(Y \neq 0)$ |
|---|---|---|---|---|---|
| $120.10^6$ | 3 | 11.37 | 9455 | .01 | .07 |

Table 1: CoCoA dataset summary statistics.

## 6.2 Evaluation Protocol for Logged Data

In order to estimate the test performance of a policy on real-world systems, off-policy evaluation is needed, as we only have access to logged exploration data. Yet, this involves in practice a number of choices and difficulties, the most documented being i) potentially infinite variance of IPS estimators (Bottou et al., 2013) and ii) propensity over-fitting (Swaminathan & Joachims, 2015a;b) where actions in the logged data can be overfitted or avoided (see Section 3.1). The former implies that it can be difficult to accurately assess the performance of new policies due to large confidence intervals, while the latter may lead to estimates that reflect large importance weights rather than rewards.

A proper evaluation protocol should therefore guard against such outcomes.

---

**Algorithm 2:** Evaluation Protocol

**Input:** $1 - \delta$: confidence of statistical test (def: 0.95); $\nu$: a max deviance ratio for effective sample size (def: 0.01);

**Output:** counter-factual estimation of $L(\pi)$ and decision to reject the null hypothesis $\{H_0$: $L(\pi) \geq L(\pi_0)\}$.

1. Split dataset $D \mapsto D^{\text{train}}, D^{\text{valid}}, D^{\text{test}}$

2. Train $\pi$ on $D^{\text{train}}$ and tune policy class and optimization hyper-parameters on $D^{\text{valid}}$ (for e.g by internal cross-validation)

3. Estimate effective sample size $n_{\text{eff}}(\pi)$ on $D^{\text{valid}}$

**if** $\frac{n_{\text{eff}}}{n} > \nu$ **then**

  Estimate $\hat{L}_{\text{SNIPS}}(\pi)$ on $D^{\text{test}}$ and test $\hat{L}_{\text{SNIPS}}(\pi) < \hat{L}(\pi_0)$ on $D^{\text{test}}$ with confidence $1 - \delta$. If the test is valid, reject $H_0$, otherwise keep it.

**else**

  Keep $H_0$, consider the estimate to be invalid.

**end**

---

A first, structuring choice is the IPS estimator. While variants of IPS exist to reduce variance, such as clipped IPS, we found Self-Normalized IPS (SNIPS, Swaminathan & Joachims, 2015b; Lefortier et al., 2016; Owen, 2013; Nedelec et al., 2017) to be more effective in practice. Indeed, it avoids the choice of a clipping threshold, generally reduces variance and is equivariant with respect to translation of the reward.

A second component is the use of importance sampling diagnostics to prevent propensity over-fitting. Lefortier et al. (2016) propose to check if the empirical average of importance weights deviates from 1. However, there is no precise guideline based on this quantity to reject estimates. Instead, we recommend to use a diagnostic on the *effective sample size* $n_{\text{eff}} = (\sum_{i=1}^{n} w_i)^2 / \sum_{i=1}^{n} w_i^2$, which measures how many samples are actually usable to perform estimation of the counterfactual estimate; we follow Owen (2013), who recommends to reject the estimate when the relative effective sample size $n_{\text{eff}}/n$ is less than 1%. A third choice is a statistical decision procedure to check if $L(\pi) < L(\pi_0)$. In theory, any statistical test against a null hypothesis $H_0$: $L(\pi) \geq L(\pi_0)$ with confidence level $1 - \delta$ can be used.

Finally, we present our protocol in Algorithm 2. Since we cannot evaluate such a protocol on purely offline data, we performed an empirical evaluation on synthetic setups where we could analytically design true

positive ($L(\pi) < L(\pi_0)$) and true negative policies. We discuss in Section 7 the concrete parameters of Algorithm 2 and their influence on false (non-)discovery rates in practice.

**Model selection with the offline protocol**  In order to make realistic evaluations, hyperparameter selection is always conducted by estimating the loss of a new policy $\pi$ in a counterfactual manner. This requires using a validation set (or cross-validation) with propensities obtained from the logging policy $\pi_0$ of the training set. Such estimates are less accurate than online ones, which would require to gather new data obtained from $\pi$, which we assume is not feasible in real-world scenarios.

To solve this issue, we have chosen to discard unreliable estimates that do not pass the effective sample size test from Algorithm 2. When doing cross-validation, it implies discarding folds that do not pass the test, and averaging estimates computed on the remaining folds. Although this induces a bias in the cross-validation procedure, we have found it to significantly reduce the variance and dramatically improve the quality of model selection when the number of samples is small, especially for the Warfarin dataset in Section 7.

## 7 Experimental Setup and Evaluation

We now provide an empirical evaluation of the various aspects of CRM addressed in this paper such as policy class modelling (CLP), estimation with soft-clipping, optimization with PPA, offline model selection and evaluation. We conduct such a study on synthetic and semi-synthetic datasets and on the real-world CoCoAdataset.

### 7.1 Experimental Validation of the Protocol

First, we study the ability of Algorithm 2 to accurately decide if a candidate policy $\pi$ is better than a reference logging policy $\pi_0$ (condition $L(\pi) \leq L(\pi_0)$) on synthetic data. Here we simulate logging policy $\pi_0$ being a lognormal distribution of known mean and variance, and an optimal policy $\pi^*$ being a Gaussian distribution. We generate a logged dataset by sampling actions $a \sim \pi_0$ and trying to evaluate policies $\hat{\pi}$ with costs observed under the logging policy. We compare the costs predicted using IPS and SNIPS offline metrics to the online metric as the setup is synthetic, it is then easy to check that indeed they are better or worse than $\pi_0$. We compare the IPS and SNIPS estimates along with their level of confidences and the influence of the effective sample size diagnostic.

Our first result is that SNIPS estimates are highly correlated to the true (online) reward (average correlation $\rho = .968$, 30% higher than IPS, see plots in Appendix D.1). Then, we have shown that with a proper choice of the $\nu$ and $\delta$ parameters, it is possible to control the False Discovery Rate (FDR) ($< 10^{-4}$) and False Non-Discovery Rate (FNDR) ($< 5.10^{-4}$) on the same setup. We observed that on simple synthetic setups the effective sample size criterion $\nu = n_{\text{eff}}/n$ is seldom necessary for policies close to the logging policy ($\pi \approx \pi_0$). For policies which were not close to the logging policy we found that standard statistical significance testing at $1 - \delta$ level was by itself not enough to guarantee a low FDR which justified the use of $\nu$. Adjusting the effective sample size can therefore influence the performance of the protocol (see Appendix D.2.2 for detailed results, D.3 for further illustrations of importance sampling diagnostics).

### 7.2 Experimental Evaluation of the Continuous Modelling and the Optimization Perspectives

In this section we introduce our empirical settings for evaluation and present our proposed CLP policy parametrization, and the influence of optimization in counterfactual risk minimization problems.

#### 7.2.1 Experimental Setup

We present the synthetic potential prediction setup, a semi-synthetic setup as well as our real-world setup.

**Synthetic potential prediction.**  We introduce simple synthetic environments with the following generative process: an unobserved random group index $g$ in $\mathcal{G}$ is drawn, which influences the drawing of a context $x$ and of an unobserved "potential" $p$ in $\mathbb{R}$, according to a joint conditional distribution $\mathcal{P}_{\mathcal{X},P|\mathcal{G}}$. Intuitively, the

potential $p$ may be compared to users a priori responsiveness to a treatment. The observed reward $-y$ is then a function of the context $x$, action $a$, and potential $p$. The causal graph corresponding to this process is given in Figure 3.

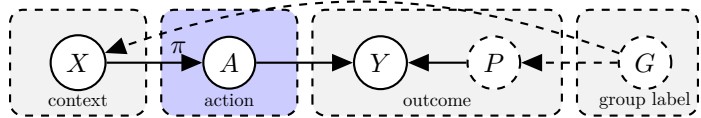

Figure 3: Causal Graph of the synthetic setting. $A$ denotes action, $X$ context, $G$ unobserved group label, $Y$ outcome and $P$ unobserved potentials. Unobserved elements are dotted.

Then, we generate three datasets ("noisymoons, noisycircles, and anisotropic", abbreviated respect. "moons, circles, and GMM" in Table 2 and illustrated in Appendix F.1, with two-dimensional contexts on 2 or 3 groups and different choices of $\mathcal{P}_{\mathcal{X},P|\mathcal{G}}$.

The goal is then to find a policy $\pi(a|x)$ that maximizes reward by adapting to an unobserved potential. For our experiments, potentials are normally distributed conditionally on the group index, $p|g \sim \mathcal{N}(\mu_g, \sigma^2)$. As many real-world applications feature a reward function that increases first with the action up to a peak and finally drops, we have chosen a piecewise linear function peaked at $a = p$ (see Appendix F.1, Figure 17), that mimics reward over the CoCoAdataset presented in Section 6. In bidding applications, a potential may represent an unknown true value for an advertisement, and the reward is then maximized when the bid (action) matches this value. In medicine, increasing drug dosage may increase treatment effectiveness but if dosage exceeds a threshold, secondary effects may appear and eclipse benefits (Barnes & Eltherington, 1966).

**Semi-synthetic setting with medical data.** We follow the setup of Kallus & Zhou (2018) using a dataset on dosage of the Warfarin blood thinner drug (War, 2009). The dataset consists of covariates about patients along with a dosage treatment prescription by a medical expert, which is a scalar value and thus makes the setting useful for continuous action modelling. While the dataset is supervised, we simulate (see details in Appendix F.1) a contextual bandit environment by using a hand-crafted reward function that is maximal for actions $a$ that are within 10% of the expert's therapeutic drug dosage, following Kallus & Zhou (2018).

**Evaluation methodology** For synthetic datasets, we generate training, validation, and test sets of size 10 000 each. For the CoCoA dataset, we consider a 50%-25%-25% training-validation-test sets. We then run each method with 5 different random intializations such that the initial policy is close to the logging policy. Hyperparameters such as the regularization parameter $\lambda$, the number of anchor points $m$, etc. are selected on a validation set with logged bandit feedback as explained in Algorithm 2. The variance $\sigma$ of the policy classes that we consider are always set to the original variance of the logging policies. We use an offline SNIPS estimate of the oat we consibtained policies, while discarding solutions deemed unsafe with the importance sampling diagnostic. On the semi-synthetic Warfarin dataset we used a cross-validation procedure to improve model selection due to the low dataset size. For estimating the final test performance and confidence intervals on synthetic and on semi-synthetic datasets, we use an online estimate by leveraging the known reward function and taking a Monte Carlo average with 100 action samples per context: this accounts for the randomness of the policy itself over given fixed samples. For offline estimates we leverage the randomness across samples to build confidence intervals: we use a 100-fold bootstrap and take percentiles of the distribution of rewards. For the CoCoA dataset, we report SNIPS estimates for the test metrics.

### 7.2.2 Empirical Evaluation

We now evaluate our proposed CLP policy parametrization and the influence of optimization in counterfactual risk minimization problems. The code to reproduce our experiments can be found at `https://github.com/criteo-research/optimization-continuous-action-crm`.

**Continuous action space requires more than naive discretization.** In Figure 4, we compare our continuous parametrization to discretization strategies that bucketize the action space and consider stochastic

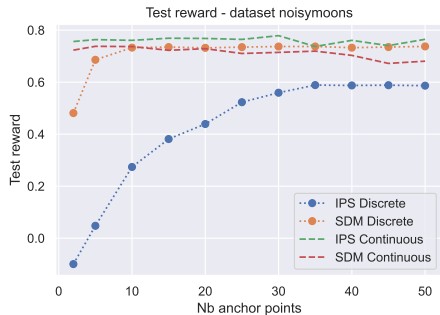

Figure 4: Continuous vs discretization strategies. Test rewards on NoisyMoons dataset with varying numbers of anchor points for our continuous parametrization for IPS and SDM, versus naive discretization with softmax policies. Note that few anchor points are sufficient to achieve good results on this dataset; this is not the case for more complicated ones (*e.g.*, Warfarin requires at least 15 anchor points).

discrete-action policies on the resulting buckets, using IPS and SDM. We add a minimal amount of noise to the deterministic DM in order to pass the $n_{\text{eff}}/n > \nu$ validation criterion, and experimented different hyperparameters and models which were selected with the offline evaluation procedure. On all synthetic datasets, the CLP continuous modeling associated to the IPS perform significantly better than discrete approaches (see also Appendix G.1), across all choices considered for the number of anchor points/buckets. To achieve a reasonable performance, naive discretization strategies require a much finer grid, and are thus also more computationally costly. The plots also show that our (stochastic) direct method strategy, where we use the same parametrization is overall outperformed by the CLP parametrization combined to IPS, highlighting a benefit of using counterfactual methods compared to a direct fit to observed rewards.

**Counterfactual cost predictor (CLP) provides a competitive parameterization for continuous-action policy learning.** We compare our CLP modelling approach to other parameterized modelings (constant, linear and non-linear described in Section 3.2) on our synthetic and semi-synthetic setups described in Section 7.2.1 as well as the CoCoAdataset presented in Section 6.1.

| | | Noisycircles | NoisyMoons | Anisotropic | Warfarin | CoCoA |
|---|---|---|---|---|---|---|
| Logging policy $\pi_0$ | | 0.5301 | 0.5301 | 0.4533 | -13.377 | 11.34 |
| scIPS | Constant | $0.6115 \pm 0.0000$ | $0.6116 \pm 0.0000$ | $0.6026 \pm 0.0000$ | $-8.964 \pm 0.001$ | $11.36 \pm 0.13$ |
| | Linear | $0.6113 \pm 0.0001$ | $0.7326 \pm 0.0001$ | $0.7638 \pm 0.0005$ | $-12.857 \pm 0.002$ | $11.35 \pm 0.02$ |
| | Poly | $0.6959 \pm 0.0001$ | $0.7281 \pm 0.0001$ | $0.7448 \pm 0.0008$ | - | $10.36 \pm 0.11$ |
| | CLP | $\mathbf{0.7674} \pm 0.0008$ | $\mathbf{0.7805} \pm 0.0004$ | $\mathbf{0.7703} \pm 0.0002$ | $\mathbf{-8.720} \pm 0.001$ | $\mathbf{11.44}^* \pm 0.10$ |
| SNIPS | Constant | $0.6115 \pm 0.0001$ | $0.6115 \pm 0.0001$ | $0.5930 \pm 0.0001$ | $-9.511 \pm 0.001$ | $11.32 \pm 0.13$ |
| | Linear | $0.6115 \pm 0.0001$ | $0.7360 \pm 0.0001$ | $0.7103 \pm 0.0003$ | $-10.583 \pm 0.005$ | $10.34 \pm 0.12$ |
| | Poly | $0.6969 \pm 0.0001$ | $0.7370 \pm 0.0001$ | $0.5801 \pm 0.0002$ | - | $11.13 \pm 0.08$ |
| | CLP | $\mathbf{0.6972} \pm 0.0001$ | $\mathbf{0.74091} \pm 0.0004$ | $\mathbf{0.7899} \pm 0.0002$ | $\mathbf{-9.161} \pm 0.001$ | $\mathbf{11.48}^* \pm 0.14$ |

Table 2: Test rewards (higher the better) for several contextual modellings (see main text for details).

In Table 2, we show a comparison of test rewards for different contextual modellings (associated to different parametric policy classes). We show the performance and the associated variance of the best policy obtained with the offline model selection procedure (Section 6.2). Specifically, we consider a grid of hyperparameters and optimized the associated CRM problem with the PPA algorithm (Section 4.2). We report here the performances of scIPS and SNIPS estimators. For the Warfarin dataset, following Kallus & Zhou (2018), we only consider the linear context parametrization baseline, since the dataset has categorical features and higher-dimensional contexts. Overall, we find our CLP parameterization to improve over all other contextual modellings, which highlights the effectiveness of the cost predictor at exploiting the continuous action structure. As all the methods here have the same sample efficiency, the superior performance of our method can be imputed to the richer policy class we use and which better models the dependency of contexts

and actions that may reduce the approximation error. We can also draw another conclusion: unlike synthetic setups, it is harder to obtain policies that beat the logging policy with large statistical significance on the CoCoAdataset where the logging policy already makes a satisfactory baseline for real-world deployment. Only CLP passes the significance test on this dataset. This corroborates the need for offline evaluation procedures, which were absent from previous works.

**Soft-clipping improves performance of the counterfactual policy learning.** Figure 5 shows the improvements in test reward of our optimization-driven strategies for the soft-clipping estimator for the synthetic datasets (see also Appendix G.3). The points correspond to different choices of the clipping parameter $M$, models and initialization, with the rest of the hyper-parameters optimized on the validation set using the offline evaluation protocol. This plot also shows that soft clipping provides benefits over hard clipping, perhaps thanks to a more favorable optimization landscape. Overall, these figures confirm that the optimization perspective is important when considering CRM problems.

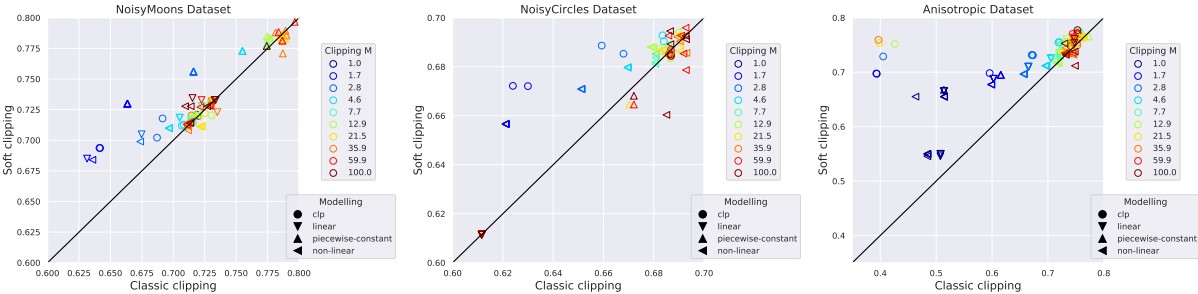

Figure 5: Influence of soft-clipping. Relative improvements in the test performance for soft- vs hard-clipping on synthetic datasets. The points correspond to different choices of the clipping parameter, models and initialization.

In Appendix G.2, we also illustrate how soft-clipping improves or competes with other clipping strategies (Metelli et al., 2021; Wang et al., 2017; Su et al., 2019).

**Proximal point algorithm (PPA) influences optimization of non-convex CRM objective functions and policy learning performance.** We illustrate in Figure 6 the improvements in test reward and in training objective of our optimization-driven strategies with the use of the proximal point algorithm (see also Appendix G.3). Here, each point compares the test metric for fixed models as well as initialization seeds, while optimizing the remaining hyperparameters on the validation set with the offline evaluation protocol. Figure 6 (left) illustrates the benefits of the proximal point method when optimizing the (non-convex) CRM objective in a wide range of hyperparameter configurations, while Figure 6 (center) shows that in many cases this improves the test reward as well. In our experiments, we have chosen L-BFGS because it was performing best among the solvers we tried (nonlinear conjugate gradient (CG) and Newton) and used 10 PPA iterations. As for computational time, for 10 PPA iterations, the computational overhead was about $3\times$ in comparison to L-BFGS without PPA. This overhead seems to be the price to pay to improve the test reward and obtain better local optima. Overall, these figures confirm that the proximal point algorithm improves performance in CRM optimization problems.

**The scIPS estimator along with CLP parametrization and PPA optimization improves upon previous state of the art methods.** We also provide a baseline comparison to stochastic direct methods, to (O-learn) of Chen et al. (2016) using their surrogate loss formulation for continous actions, to kernel smoothing (KS) (Kallus & Zhou, 2018) who propose a counterfactual method using kernel density estimation. Their approach is based on an automatic kernel bandwidth selection procedure which did not perform well on our datasets except Warfarin; instead, we select the best bandwidth on a grid through cross-validation and selecting it through our offline protocol. We also investigate their self-normalized (SN) variant, which is presented in their paper but not used in their experiments; it turned out to have lower performances in practice. Moreover, we experimented using the generic doubly robust method from Demirer et al. (2019) but

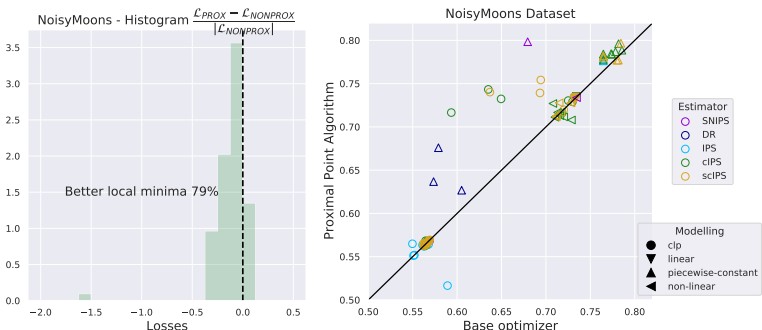

Figure 6: Influence of proximal point optimization. Relative improvements in the training objective w and w/o using the proximal point method (left) and relative improvements in the test performance w and w/o using the proximal point method (right).

could not reach satisfactory results using the parameters and feature maps that were used in their empirical section and with the specific closed form estimators for their applications. Nevertheless, by adapting their method with more elaborated models and feature maps, we managed to obtain performances beating the logging policy; these modifications would make promising directions for future research venues. We eventually also compare to (CATS) (Majzoubi et al., 2020) who propose an offline variant of their contextual bandits algorithm for continuous actions. We used the code of the authors and obtained poor performances with their offline variant. We do not provide results on their method for the CoCoAdataset as we could not access the logged propensities to use the offline evaluation protocol. For the SDM on CoCoA, we did not manage to simultaneously pass the ESS diagnostic and achieve statistical significance, probably due to the noise and variance of the dataset.

| | Noisycircles | NoisyMoons | Anisotropic | Warfarin | CoCoA |
|---|---|---|---|---|---|
| Stochastic Direct Method | $0.6205 \pm 0.0004$ | $0.7225 \pm 0.0006$ | $0.6383 \pm 0.0003$ | $-9.714 \pm 0.013$ | - |
| O-learn | $0.608 \pm 0.0002$ | $0.645 \pm 0.0003$ | $0.754 \pm 0.0002$ | $-9.407 \pm 0.004$ | $11.03 \pm 0.15$ |
| KS | $0.612 \pm 0.0001$ | $0.734 \pm 0.0001$ | $0.785 \pm 0.0002$ | $-10.19^*$ | $11.38 \pm 0.07$ |
| SN-KS | $0.609 \pm 0.0001$ | $0.595 \pm 0.0001$ | $0.652 \pm 0.0001$ | $-12.569 \pm 0.001$ | $9.14 \pm 0.94$ |
| CATS offline | $0.589 \pm 0.0011$ | $0.592 \pm 0.0011$ | $0.569 \pm 0.0012$ | $-12.236 \pm 0.2548$ | - |
| Ours | $\mathbf{0.767} \pm 0.0008$ | $\mathbf{0.781} \pm 0.0004$ | $\mathbf{0.770} \pm 0.0002$ | $\mathbf{-8.720} \pm 0.001$ | $\mathbf{11.44}^* \pm 0.10$ |

Table 3: Test rewards (higher the better) for previous methods for the logged bandit problem with continuous actions

# 8 Conclusion and Discussion

In this paper, we address the problem of counterfactual learning of stochastic policies on real data with continuous actions. This raises several challenges about different steps of the CRM pipeline such as (i) modelization, (ii) optimization, and (iii) evaluation. First, we propose a new parametrization based on a joint kernel embedding of contexts and actions, showing competitive performance. Second, we underline the importance of optimization in CRM formulations with soft-clipping and proximal point methods. We provide statistical guarantees of our estimator and the policy class we introduced. Third, we propose an offline evaluation protocol and a new large-scale dataset, which, to the best of our knowledge, is the first with real-world logged propensities and continuous actions. For future research directions, we would like to investigate the doubly robust estimator with the CLP parametrization of stochastic policies with continuous actions: this estimator achieves the best results in the discrete action case but requires further techniques to be adapted to continuous actions.

**Acknowledgments**

HZ acknowledges support from the Gatsby Charitable Foundation. HZ also acknowledges support by the KARAIB AI chair (ANR-20-CHIA-0025-01) and the H2020 Research Infrastructures Grant EBRAIN-Health 101058516. JM was supported by ERC grant number 101087696 (APHELEIA project) and by ANR 3IA MIAI@Grenoble Alpes (ANR-19-P3IA-0003).

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

## A  Review of Off-policy Estimators

*Clipped estimator.* In Equation (3), the counterfactual approach tackles the distribution mismatch between the logging policy $\pi_0(\cdot|x)$ and the evaluated policy $\pi$ in $\Pi$ via importance sampling and uses inverse propensity scoring (IPS, Horvitz & Thompson (1952)). However, the empirical (IPS) estimator has large variance and may overfit negative feedback values $y_i$ for samples that are unlikely under $\pi_0$ (see motivation for clipped estimators in B.1), resulting in higher variances. Clipping the importance sampling weights in Eq. (16) as Bottou et al. (2013) mitigates this problem, leading to a clipped (cIPS) estimator

$$\hat{L}_{\text{cIPS}}(\pi) = \frac{1}{n} \sum_{i=1}^{n} y_i \min \left\{ \frac{\pi(a_i|x_i)}{\pi_{0,i}}, M \right\}. \tag{16}$$

Smaller values of $M$ reduce the variance of $\hat{L}(\pi)$ but induce a larger bias. Swaminathan & Joachims (2015a) also propose adding an empirical variance penalty term controlled by a factor $\lambda$ to the empirical risk $\hat{L}(\pi)$. Specifically, they write $\nu_i(\pi) = y_i \min \left( \frac{\pi(a_i|x_i)}{\pi_0(a_i|x_i)}, M \right)$ and consider the empirical variance for regularization:

$$\hat{V}_{\text{cIPS}}(\pi) = \frac{1}{n-1} \sum_{i=1}^{n} (\nu_i(\pi) - \bar{\nu}(\pi))^2, \qquad \text{with} \quad \bar{\nu}(\pi) = \frac{1}{n} \sum_{i=1}^{n} \bar{\nu}_i(\pi), \tag{17}$$

which is subsequently used to obtain a regularized objective $\mathcal{L}$ with hyperparameters $M$ for clipping and $\lambda$ for variance penalization, respectively, so that $\Omega_{\text{cIPS}}(\pi) = \lambda \sqrt{\frac{\hat{V}_{\text{cIPS}}(\pi)}{n}}$ and:

$$\mathcal{L}(\pi) = \hat{L}_{\text{cIPS}}(\pi) + \lambda \sqrt{\frac{\hat{V}_{\text{cIPS}}(\pi)}{n}}. \tag{18}$$

*The self-normalized estimator.* Swaminathan & Joachims (2015b) also introduce a regularization mechanism for tackling the so-called *propensity overfitting* issue, occuring with rich policy classes, where the method would focus only on maximizing (resp. minimizing) the sum of ratios $\pi(a_i|x_i)/\pi_{0,i}$ for negative (resp. positive) costs. This effect is corrected through the following *self-normalized importance sampling* (SNIPS) estimator (Owen, 2013, see also), which is equivariant to additive shifts in cost values:

$$\hat{L}_{\text{SNIPS}}(\pi) = \frac{\sum_{i=1}^{n} y_i w_i^{\pi}}{\sum_{i=1}^{n} w_i^{\pi}}, \quad \text{with} \quad w_i^{\pi} = \frac{\pi(a_i|x_i)}{\pi_{0,i}}. \tag{19}$$

The SNIPS estimator is also associated to $\Omega_{\text{SNIPS}}(\pi) = \lambda \sqrt{\frac{\hat{V}_{\text{SNIPS}}(\pi)}{n}}$ which uses an empirical variance estimator that writes as:

$$\hat{V}_{\text{SNIPS}}(\pi) = \frac{\sum_{i=1}^{n} \left( w_i^{\pi} \left( y_i - \hat{L}_{\text{SNIPS}}(\pi) \right) \right)^2}{\left( \sum_{i=1}^{n} w_i^{\pi} \right)^2}. \tag{20}$$

*Direct methods.* Such direct methods (DM) fit the loss values over contexts and actions in observed data with an estimator $\hat{\eta}(x, a)$, for instance by using ridge regression to fit $y_i \approx \hat{\eta}(x_i, a_i)$, and to then use the deterministic greedy policy $\hat{\pi}_{\mathrm{DM}}(x) = \arg\min_a \hat{\eta}(x, a)$. These may suffer from large bias since it focuses on estimating losses mainly near actions that appear in the logged data but have the benefit of avoiding the high-variance problems of IPS-based methods. While such greedy deterministic policies may be sufficient for exploitation, stochastic policies may be needed in some situations, for instance when one wants to still encourage some exploration in a future round of data logs. Using a stochastic policy also allows us to obtain more accurate off-policy estimates when performing cross-validation on logged data. Then, it may be possible to define a stochastic version of the direct method by adding Gaussian noise with variance $\sigma^2$:

$$\hat{\pi}_{\mathrm{SDM}}(\cdot|x) = \mathcal{N}(\hat{\pi}_{\mathrm{DM}}(x), \sigma^2), \tag{21}$$

In the context of offline evaluation on bandit data, such a smoothing procedure may also be seen as a form of kernel smoothing for better estimation (Kallus & Zhou, 2018).

*Doubly robust estimators.* Additionally, such direct loss estimators can be effective when few samples are available, and may be combined with IPS estimators in the so-called doubly robust estimator (DR, see, e.g. Dudik et al. (2011)). This approach consists of correcting the bias of the DM estimator by applying IPS to the residuals $y_i - \hat{\eta}(x_i, a_i)$, thus using $\hat{\eta}$ as a control variate to decrease the variance of IPS. For discrete actions, the DR estimator takes the form

$$\hat{L}_{\mathrm{DR}}(\pi) = \frac{1}{n} \sum_{i=1}^{n} \frac{\pi(a_i|x_i)}{\pi_{0,i}} \left( y_i - \hat{\eta}(x_i, a_i) \right) + \frac{1}{n} \sum_{i=1}^{n} \sum_{a \in \mathcal{A}} \pi(a|x_i)\hat{\eta}(x_i, a).$$

We provide a more in depth discussion on a doubly robust adaptation of our estimator in Appendix G.4.

## B  Motivation for Clipped Estimators

In this section we provide an illustration of the logarithmic soft clipping and a motivation example for clipping strategies in counterfactual systems.

### B.1  A Toy Example for Soft Clipping

Here we provide a motivation example for clipping strategies in counterfactual systems in a toy example. In Figure 7 we provide an example of large variance and loss overfitting problem.

We recall the data generation: a hidden group label $g$ in $\mathcal{G}$ is drawn, and influences the associated context distribution $x$ and of an unobserved potential $p$ in $\mathbb{R}$, according to a joint conditional distribution $P_{X,P|G}$ The observed reward $r$ is then a function of the context $x$, action $a$, and potential $p$. Here, we design one outlier (big red dark dot on Figure 7 left). This point has a noisy reward $r$, higher than neighbors, and a potential $p$ high as its neighbors have a low potential. We artificially added a noise in the reward function $f$ that can be written as:

$$r = f(a, x, p) + \varepsilon, \quad \varepsilon \sim \mathcal{N}(0, 1)$$

As explained in Section 7.2.1, the reward function is a linear function, with its maximum localized at the point $x = p(x)$, i.e. at the potential sampled. The observability of the potential $p$ is only through this reward function $f$. Hereafter, we compare the optimal policy computed, using different types of estimators.

The task is to predict the high potentials (red circles) and low potentials (blue circles) in the ground truth data (left). Unfortunately, a rare event sample with high potential is put in the low potential cluster (big dark red dot). The action taken by the logging policy is low while the reward is high: this sample is an outlier because it has a high reward while being a high potential that has been predicted with a low action. The resulting unclipped estimator is biased and overfits this high reward/low propensity sample. The rewards of the points around this outlier are low as the diameter of the points in the middle figure show. Inversely, clipped estimator with soft-clipping succeeds to learn the potential distributions, does not overfit the outlier, and has larger rewards than the clipping policies as the diameter of the points show.

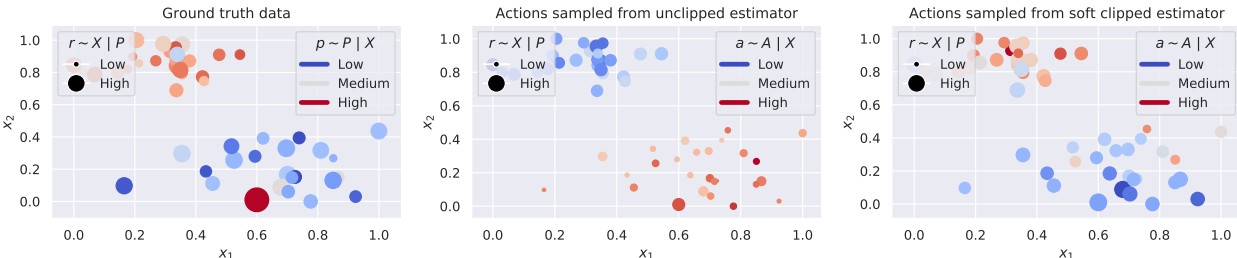

Figure 7: High variance and loss overfitting. Unlikely $(\pi_{0,i} \approx 0)$ sample $(x_1, x_2) = (0.6, 0.)$ with high reward $r$ (left) results in larger variance and loss overfitting for the unclipped estimator (middle) unlike clipped estimator (right).

## C  Motivation for Counterfactual Methods

Direct methods (DM) learns a reward/cost predictor over the joint context-action space $\mathcal{X} \times \mathcal{A}$ but ignore the potential mismatch between the evaluated policy and the logging policy and $\pi_0$. When the logged data does not cover the joint context-action space $\mathcal{X} \times \mathcal{A}$ sufficiently, direct methods rather fit the region where the data has been sampled and may therefore lead to overfitting (Bottou et al., 2013; Dudik et al., 2011; Swaminathan & Joachims, 2015b). Counterfactual methods instead learn probability distributions directly with a re-weighting procedure which allow them to fit the context-action space even with fewer samples.

In this toy setting we aim to illustrate this phenomenon for the DM and the counterfactual method. We create a synthetic 'Chess' environment of uni-dimensional contexts and actions where the logging policy purposely covers only a small area of the action space, as illustrated in Fig. 8. The reward function is either 0, 0.5 or 1 in some areas which follow a chess pattern. We use a lognormal logging policy which is peaked in low action values but still has a common support with the policies we optimize using the CRM or the DM.

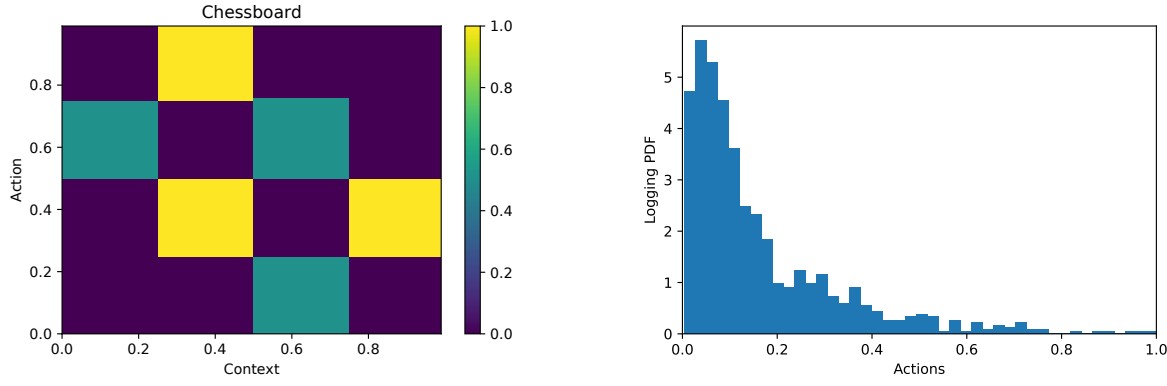

Figure 8: 'Chess' toy synthetic setting (left) and lognormal logging policy (right).

Having set that environment and the logging policy, we illustrate in Fig. 9 the logging dataset, the actions sampled by the policy learned by a Direct Method and eventually the actions sampled by a counterfactual IPS estimator. To assess a fair comparison between the two methods, we use the same continuous action modelling with the same parameters (CLP parametrization with $m = 5$ anchor points).

This toy example illustrate the mentioned phenomenon in how the counterfactual estimator learns a re-balanced distribution that maps the contexts to the actions generating higher rewards than the DM. The latter only learns a mapping that is close to the actions sampled by the logging and only covers a smaller set of actions.

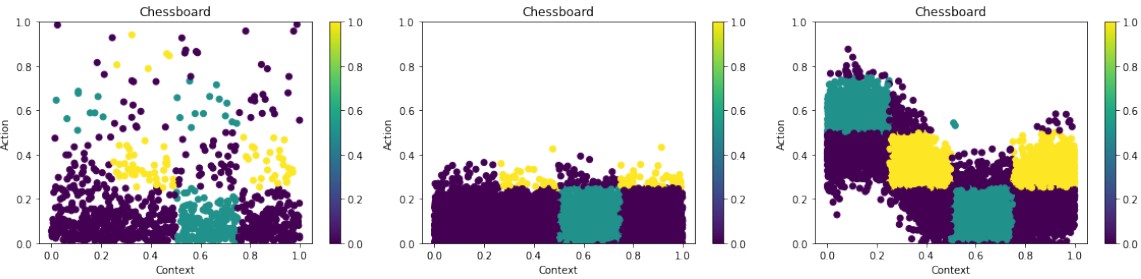

Figure 9: Logged data (left), action sampled by direct method (middle) and action sampled by counterfactual policy.

# D    Motivation for the Offline Evaluation Protocol

In this part we demonstrate the offline/online correlation of the estimator we use for real-world systems and for validation of our methods even in synthetic and semi-synthetic setups. We provide further explanations of the necessity of importance sampling diagnostics and we perform experiments to empirically assess the rate of false discoveries of our protocol.

## D.1    Correlation of Self-Normalized Importance Sampling with Online Rewards

We show in Figures 10,12,11 comparisons of IPS and SNIPS against an on-policy estimate of the reward for policies obtained from our experiments for linear and non-linear contextual modellings on the synthetic datasets, where policies can be directly evaluated online. Each point represents an experiment for a model and a hyperparameter combination. We measure the $R^2$ score to assess the quality of the estimation, and find that the SNIPS estimator is indeed more robust and gives a better fit to the on-policy estimate. Note also that overall the IPS estimates illustrate severe variance compared to their SNIPS estimate. While SNIPS indeed reduces the variance of the estimate, the bias it introduces does not deteriorate too much its (positive) correlation with the online evaluation.

These figures further justify the choice of the self-normalized estimator SNIPS (Swaminathan & Joachims, 2015b) for offline evaluation and validation to estimate the reward on held-out logged bandit data. The SNIPS estimator is indeed more robust to the reward distribution thanks to its equivariance property to additive shifts and does not require hyperparameter tuning.

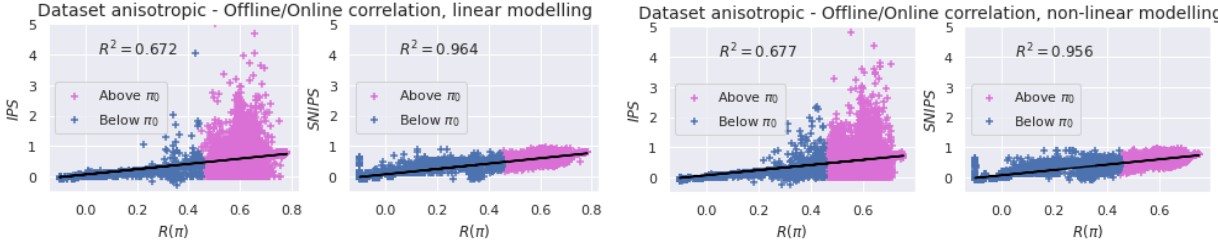

Figure 10: **Correlation between offline and online estimates on Anisotropic synthetic data**. Linear (left) and non-linear (right) contextual modellings. Ideal fit would be $y = x$.

## D.2    Experimental validation of the protocol

We empirically evaluate our offline evaluation protocol on a toy setting where we simulate a logging policy $\pi_0$ being a lognormal distribution of known mean and variance, and an optimal policy $\pi^*$ being a Gaussian distribution. We generate a logged dataset by sampling actions $a \sim \pi_0$ and trying to evaluate policies $\hat{\pi}$ with

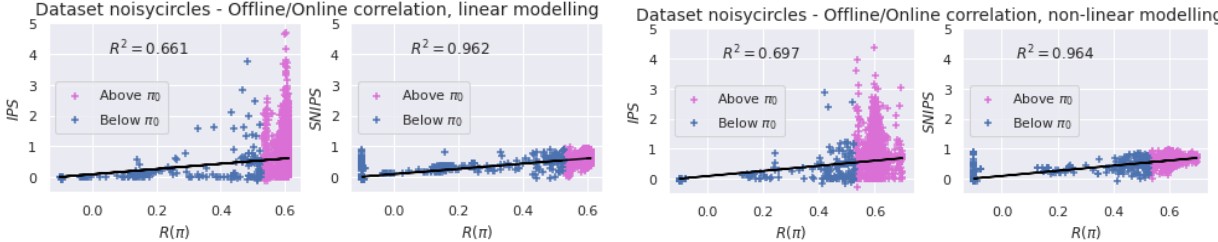

Figure 11: **Correlation between offline and online estimates on NoisyCircles synthetic data**. Linear (left) and non-linear (right) contextual modellings. Ideal fit would be $y = x$.

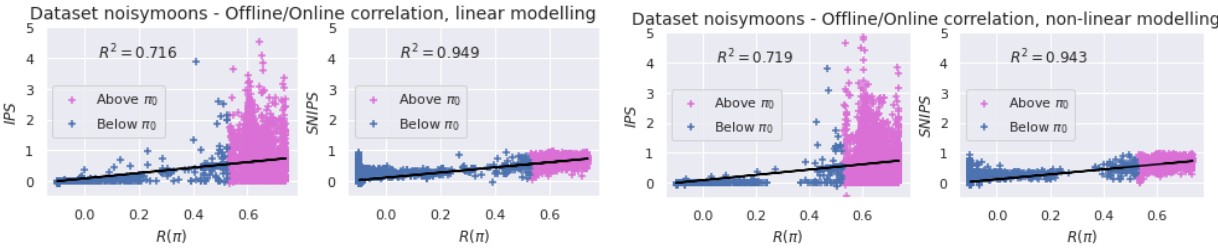

Figure 12: **Correlation between offline and online estimates on NoisyMoons synthetic data**. Linear (left) and non-linear (right) contextual modellings. Ideal fit would be $y = x$.

costs observed under the logging policy. We compare the costs predicted using IPS and SNIPS offline metrics to the online metric as the setup is synthetic, it is then easy to check that indeed they are better or worse than $\pi_0$. We compare the IPS and SNIPS estimates along with their level of confidences and the influence of the effective sample size diagnostic. Offline evaluations of policies $\hat{\pi}$ illustrated in Figure 13 are estimated from logged data $(x_i, a_i, y_i, \pi_0)_{i=1...n}$ where $a_i \sim \pi_0(\cdot|x_i)$ and where the policy risk would be optimal under the oracle policy $\pi^*$.

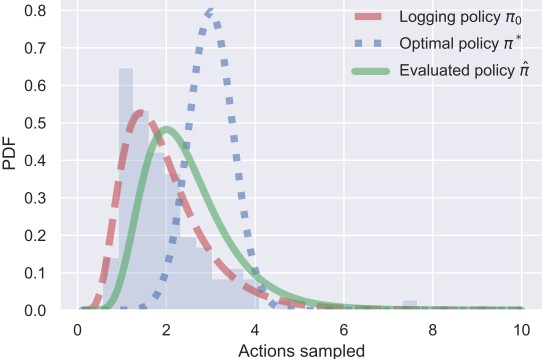

Figure 13: Illustration of policies: logging policy $\pi_0$, optimal $\pi^*$ and example policy $\hat{\pi}$.

While the goal of counterfactual learning is to find a policy $\hat{\pi}$ which is as close a possible to the optimal policy $\pi^*$, based on samples drawn from a logging policy $\pi_0$, it is in practice hard to assess the statistical significance of a policy that is too "far" from the logging policy. Offline importance sampling estimates are indeed limited when the distribution mismatch between the evaluated policy and the logging policy (in terms of KL divergence $D_{KL}(\pi_0||\hat{\pi})$) is large. Therefore we create a setup where we evaluate the quality of offline estimates for policies (i) "close" to the logging policy (meaning the KL divergence $D_{KL}(\pi_0||\hat{\pi})$ is low) and

(ii) "close" to the oracle optimal policy (meaning the KL divergence $D_{KL}(\pi^*||\hat\pi)$ is low). In this experiment, we focus on evaluating the ability of the offline protocol to correctly assess whether $L(\hat\pi) \leq L(\pi_0)$ or not by comparing to online truth estimates. Specifically, for both setups (i) and (ii), we compare the number of False Positives (FP) and False Negatives (FN) of the two offline protocols for $N = 2000$ initializations, by adding Gaussian noise to the parameters of the closed form policies. False negatives are generated when the offline protocol keeps $H_0 : L(\pi) \geq L(\pi_0)$ while the online evaluation reveals that $L(\pi) \leq L(\pi_0)$, while false positives are generated in the opposite case when the protocol rejects $H_0$ while it is true. We also show histograms of the differences between online and offline boundary decisions for $(L(\hat\pi) < L(\pi_0))$, using bootstrapped distribution of SNIPS estimates to build confidence intervals.

### D.2.1 Validation of the use of SNIPS estimates for the offline protocol

To assess the performance of our evaluation protocol, we first compare the use of IPS and SNIPS estimates for the offline evaluation protocol and discard solutions with low importance sampling diagnostics $\frac{n_{\text{eff}}}{n} < \nu$ with the recommended value $\nu = 0.01$ from Owen (2013). In Table 4, we provide an analysis of false positives and false negatives in both setups. We first observe that for setup (i) the SNIPS estimates has both fewer false positives and false negatives. Note that is setup is probably more realistic for real-world applications where we want to ensure incremental gains over the logging policy. In setup (ii) where importance sampling is more likely to fail when the evaluated policy is too "far" from the logging policy, we observe that the SNIPS estimate has a drastically lower number of false negatives than the IPS estimate, though it slightly has more false positives, thus illustrating how conservative this estimator is.

| Offline Protocol | | Setup (i) | | | | Setup (ii) | | | |
|---|---|---|---|---|---|---|---|---|---|
| | | IPS | | SNIPS | | IPS | | SNIPS | |
| | | $\hat\pi \succeq \pi_0$ | Keep $H_0$ | $\hat\pi \succeq \pi_0$ | Keep $H_0$ | $\hat\pi \succeq \pi_0$ | Keep $H_0$ | $\hat\pi \succeq \pi_0$ | Keep $H_0$ |
| "Truth" | $\hat\pi \succeq \pi_0$ | 1282 | 24 | 1296 | **10** | 1565 | 67 | 1631 | **1** |
| | Keep $H_0$ | 19 | 675 | **0** | 694 | **0** | 368 | 6 | 362 |

Table 4: Comparison of false positives and false negatives: Perturbation to the logging policy $\pi_0$ (setup (i)) and perturbation to the optimal policy (setup (ii)). The SNIPS estimator yields less FN and FP on setup (i), while being more effective on setup (ii) as well by inducing a drastically lower FP rate than IPS and a low FN rate. The effective sample size threshold is fixed at $\nu = 0.01$

We then provide in Fig. 14 histograms of the differences of the upper boundary decisions between online estimates and bootstrapped offline estimates over all samples for both setups (i, left) and (ii, right). Both histograms illustrate how the IPS estimate underestimates the value of the reward with regard to the online estimate, unlike the SNIPS estimates. In the setup (ii) in particular, the IPS estimate underestimates severely the reward, which may explain why IPS has lower number of false positives when the evaluated policy is far from the logging policy. However in both setups, IPS has a higher number of false negatives. We also observed that our SNIPS estimates were highly correlated to the true (online) reward (average correlation $\xi = .968$, 30% higher than IPS, see plots in Appendix D.1) for the synthetic setups presented in section 7.2.1, which therefore confirms our findings.

### D.2.2 Influence of the effective sample size criteria in the evaluation protocol

In this setup we vary the effective sample size (ESS) threshold and show in Fig. 15 how it influences the performance of the offline evaluation protocol for the two previously discussed setups where we consider evaluations of (i) perturbations of the logging policy (left) and (ii) perturbations of the optimal policy (right) in our synthetic setup. We compute precision, recall and F1 scores for each threshold values between 0 and 1. One can see that for low threshold values where no policies are filtered, precision, recall and F1 scores remain unchanged. Once the ESS raises above a certain threshold, undesirable policies start being filtered but more false negatives are created when the ESS is too high. Overall, ESS criterion is relevant for both setups. However, we observe that on simple synthetic setups the effective sample size criterion $\nu = n_{\text{eff}}/n$ is seldom necessary for policies close to the logging policy ($\pi \approx \pi_0$). Conversely, for policies which are not close to the logging policy the standard statistical significance testing at $1 - \delta$ level was by itself not enough

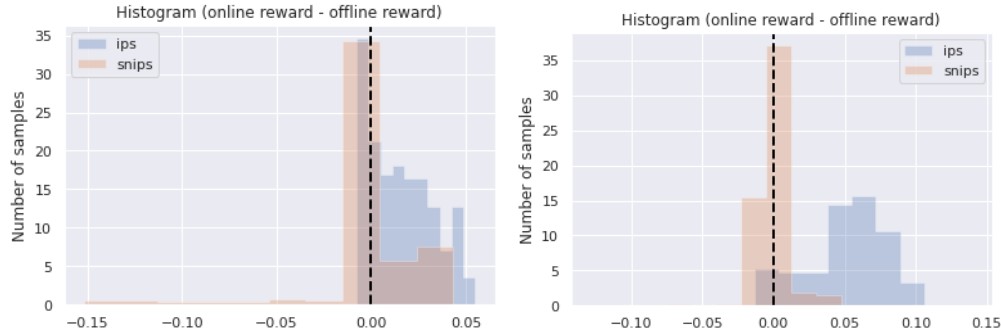

Figure 14: **Histogram of differences between online reward and offline lower confidence bound.** Perturbation to the logging policy $\pi_0$ (left), perturbation to the optimal policy $\pi^*$ (right). Effective sample size threshold $\nu = 0.01$

to guarantee a low false discovery rate (FDR) which justified the use of $\nu$. Adjusting the effective sample size can therefore influence the performance of the protocol (see Appendix D.3 for further illustrations of importance sampling diagnostics in what-if simulations).

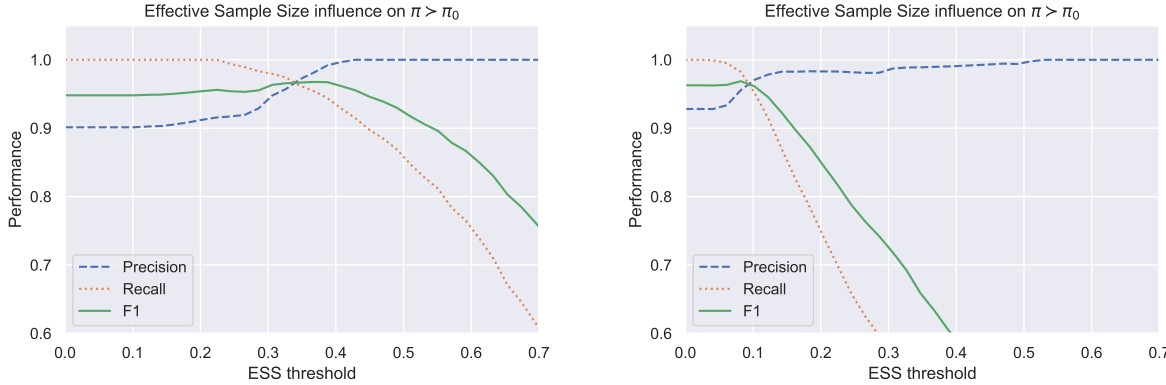

Figure 15: **Precision, recall and F1 score varying with the ESS threshold on synthetic setups (i) and (ii).** Setup (i) perturbation of the logging policy (left) and setup (ii) perturbation to the optimal policy (right). The ESS threshold can maximize the F1 score.

### D.3   Importance Sampling Diagnostics in What-If simulations

Importance sampling estimators rely on weighted observations to address the distribution mismatch for offline evaluation, which may cause large variance of the estimator. Notably, when the evaluated policy differs too much from the logging policy, many importance weights are large and the estimator is inaccurate. We provide here a motivating example to illustrate the effect of importance sampling diagnostics in a simple scenario.

When evaluating with SNIPS, we consider an "effective sample size" quantity given in terms of the importance weights $w_i = \pi(a_i|x_i)/\pi_0(a_i|x_i)$ by $n_e = (\sum_{i=1}^n w_i)^2 / \sum_{i=1}^n w_i^2$. When this quantity is much smaller than the sample size $n$, this indicates that only few of the examples contribute to the estimate, so that the obtained value is likely a poor estimate. Apart from that, we note also that IPS weights have an expectation of 1 when summed over the logging policy distribution (that is $\mathbb{E}_{(x,a)\sim\pi_0}[\pi(a|x)/\pi_0(a|x)] = 1$.). Therefore, another sanity check, which is valid for any estimator, is to look for the empirical mean $1/n\sum_{i=1}^n \pi(a_i|x_i)/\pi_{0,i}$ and compare its deviation to 1. In the example below, we illustrate three diagnostics: (i) the one based on effective sample size described in Section 6; (ii) confidence intervals, and (iii) empirical mean of IPS weights. The three of them coincide and allow us to remove test estimates when the diagnostics fail.

**Example D.1.** *What-if simulation: For $x$ in $\mathbb{R}^d$, let $\max(x) = \max_{1 \leq j \leq d} x_j$; we wish to estimate $\mathbb{E}(\max(X))$ for $X$ i.i.d $\sim \pi_\mu = \mathcal{N}(\mu, \sigma)$ where samples are drawn from a logging policy $\pi_0 = \log \mathcal{N}(\lambda_0, \sigma_0)$ ($d = 3, (\lambda_0, \sigma_0) = (1, 1/2)$) and analyze parameters $\mu$ around the mode of the logging policy $\mu_{\pi_0}$ with fixed variance $\sigma = 1/2$. In this parametrized policy example, we see in Fig. 16 that $n_e/n \ll 1$, confidence interval range increases and $\sum_{i=1}^{n} \frac{\pi(a_i|x_i)}{\pi_{0,i}} \neq 1$ when the parameter $\mu$ of the policy being evaluated is far away from the logging policy mode $\mu_{\lambda_0}$.*

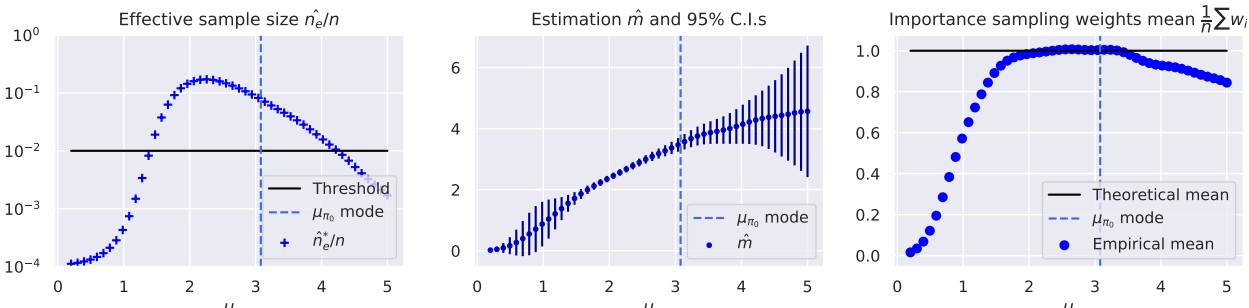

Figure 16: **Importance sampling diagnostics**. Ideal importance sampling: i) effective sample $n_e/n$ close to 1, ii) low confidence intervals (C.I.s) for $\hat{m}$, iii) empirical mean $\frac{1}{n} \sum_i w_i$ close to 1. Note that when $\mu$ differs too much from $\mu_{\pi_0}$, importance sampling fails.

Note that in this example, the parameterized distribution that is learned (multivariate Gaussian) is not the same as the parameterized distribution of the logging policy (multivariate Lognormal) which skewness may explain the asymmetry of the plots. This points out another practical problem: even though different parametrization of policies is theoretically possible, the probability density masses overlap is in practice what is most important to ensure successful importance sampling. This observation is of utmost interest for real-life applications where the initialization of a policy to be learned needs to be "close" to the logging policy; otherwise importance sampling may fail from the very first iteration of an optimization in learning problems.

# E  Analysis of the Excess Risk

In this appendix, we provide details and proofs on the excess risk guarantees that are given in Section 4 and 5. We start by recalling the definitions of an $\varepsilon$ covering and the one of our soft-clipping operator $\zeta$ provided in Eq. (11).

**Definition E.1** (epsilon covering and metric entropy). *An $\varepsilon$-covering is a subset $A_0 \subseteq A$ such that $A$ is contained in the union of balls of radius $\varepsilon$ centered in points in $A_0$, in the metric induced by a norm $\|\cdot\|$. The cardinality of the smallest $\varepsilon$-covering is denoted by $\mathcal{H}(\varepsilon, A, \|\cdot\|)$ and its logarithm is called the metric entropy.*

For any threshold parameter $M \geq 0$ and importance weight $w \geq 0$, the soft-clip operator $\zeta$ is defined by

$$\zeta(w, M) = \begin{cases} w & \text{if } w \leq M \\ \alpha(M) \log(w + \alpha(M) - M) & \text{otherwise} \end{cases},$$

where $\alpha(M)$ is such that $\alpha(M) \log(\alpha(M)) = M$.

## E.1  Ommitted Proofs

We start by defining our complexity measure $C_n(\Pi, M)$, which will be upper-bounded by the metric entropy in sup-norm at level $\varepsilon = 1/n$ of the following function set,

$$\mathcal{F}_{\Pi,M} := \left\{ f_\pi : (x, a, y) \mapsto 1 + \frac{y}{S} \zeta\left(\frac{\pi(a|x)}{\pi_0(a|x)}, M\right) \quad \text{for some } \pi \in \Pi \right\}, \tag{22}$$

where $S = \zeta(W, M)$. The function set corresponds to clipped prediction errors of policies $\pi$ normalized into $[0, 1]$. More precisely, to define rigorously $C_n(\Pi, M)$, we denote for any $n \geq 1$ and $\varepsilon > 0$, the complexity of a class $\mathcal{F}$ by

$$\mathcal{H}_\infty(\varepsilon, \mathcal{F}, n) = \sup_{(x_i, a_i, y_i) \in (\mathcal{X} \times \mathcal{A} \times \mathcal{Y})^n} \mathcal{H}(\varepsilon, \mathcal{F}(\{x_i, a_i, y_i\}), \|\cdot\|_\infty), \tag{23}$$

where $\mathcal{F}(\{x_i, a_i, y_i\}) = \{(f(x_1, a_1, y_1), \ldots, f(x_n, a_n, y_n)), f \in \mathcal{F}\} \subseteq \mathbb{R}^n$. Then, $C_n(\Pi, M)$ is defined by

$$C_n(\Pi, M) = \log \mathcal{H}_\infty(1/n, \mathcal{F}_{\Pi, M}, 2n). \tag{24}$$

We are now ready to prove Proposition 4.1 that we restate below.

**Proposition 4.1** (Generalization bound for $\hat{L}_{\text{scIPS}}(\pi)$). *Let $\Pi$ be a policy class and $\pi_0$ a logging policy, under which an input-action-cost triple follows $\mathcal{D}_{\pi_0}$. Assume that $-1 \leq y \leq 0$ a.s. when $(x, a, y) \sim \mathcal{D}_{\pi_0}$ and that the importance weights are bounded by $W$. Then, with probability at least $1 - \delta$, the IPS estimator with soft clipping (12) on $n$ samples satisfies*

$$\forall \pi \in \Pi, \qquad L(\pi) \leq \hat{L}_{\text{scIPS}}(\pi) + O\left(\sqrt{\frac{\hat{V}_{\text{scIPS}}(\pi)\left(C_n(\Pi, M) + \log\frac{1}{\delta}\right)}{n}} + \frac{S\left(C_n(\Pi, M) + \log\frac{1}{\delta}\right)}{n}\right),$$

*where $S = \zeta(W, M)$, $\hat{V}_{\text{scIPS}}(\pi)$ denotes the empirical variance of the cost estimates (20), and $C_n(\Pi, M)$ is a complexity measure (24) of the policy class.*

*Proof.* Let $\Pi$ be a policy class, $\pi_0$ be a logging policy, and $\delta > 0$. Let $M \geq 0$ be a threshold parameter, $W \geq \sup_{a,x}\{\pi(a|x)/\pi_0(a|x)\} \geq 0$ a bound on the importance weights, and $S = \zeta(W, M)$.

Let first consider the finite setting, in which case $C_n(\Pi, M) \leq \log|\Pi|$. Since all functions in $\mathcal{F}_{\Pi, M}$ defined in Eq. (22) take values in $[0, 1]$, we can apply the concentration bound of Maurer & Pontil (2009, Corollary 5) to $\mathcal{F}_{\Pi, M}$, which yields that with probability at least $1 - \delta$, for any $\pi \in \Pi$

$$\mathbb{E}_{(x,a,y) \sim \mathcal{D}_{\pi_0}}[f_\pi(x, a, y)] - \frac{1}{n}\sum_{i=1}^n f_\pi(x_i, a_i, y_i) \leq \sqrt{\frac{2\hat{V}_{\text{scIPS}}(\pi)\log(2|\Pi|/\delta)}{n}} + \frac{7\log(2|\Pi|/\delta)}{3(n-1)}, \tag{25}$$

where $\hat{V}_{\text{scIPS}}(\pi)$ is the sample variance defined in (13). Furthermore, note that by construction of the $f_\pi$, for any $\pi \in \Pi$,

$$\mathbb{E}_{(x,a,y) \sim \mathcal{D}_{\pi_0}}[f_\pi(x, a, y)] = 1 + \frac{L^M(\pi)}{S} \quad \text{and} \quad \frac{1}{n}\sum_{i=1}^n f_\pi(x_i, a_i, y_i) = 1 + \frac{\hat{L}_{\text{scIPS}}(\pi)}{S},$$

where $L^M(\pi) = \mathbb{E}_{(x,a,y) \sim \mathcal{D}_{\pi_0}}\left[y\zeta\left(\pi(a|x)/\pi_0(a|x), M\right)\right]$ denotes the clipped expected risk of the policy $\pi$ and $\hat{L}_{\text{scIPS}}(\pi)$ is defined in (12). Thus, multiplying (25) by $S$ and using that $L(\pi) \leq L^M(\pi)$ (since $y \leq 0$ and $\zeta(w, M) \leq w$ for all $w$), we get that with probability $1 - \delta$,

$$L(\pi) \leq \hat{L}_{\text{scIPS}}(\pi) + \sqrt{\frac{2\hat{V}_{\text{scIPS}}(\pi)\log(2|\Pi|/\delta)}{n}} + S\frac{7\log(2|\Pi|/\delta)}{3(n-1)}, \qquad \forall \pi \in \Pi.$$

The finite setting may finally be extended to infinite policy classes by leveraging Maurer & Pontil (2009, Theorem 6) as in (Swaminathan & Joachims, 2015a). This essentially consists in replacing $|\Pi|$ above with an empirical $\ell_\infty$ covering number of $\mathcal{F}_{\Pi, M}$ of size $\mathcal{H}_\infty(1/n, \mathcal{F}_{\Pi, M}, 2n)$. Note that the number of empirical samples $2n$ is due to the double-sample method used by Maurer & Pontil (2009). $\qquad\square$

We now state the excess risk upper-bound Proposition E.1 and provide the proof. The following proposition is an intermediate result that will allow us to derive the Proposition 5.1.

**Proposition E.1.** *Consider the notations and assumptions of Proposition 4.1. Let $\hat{\pi}^{CRM}$ be the solution of the CRM problem in Eq. (15) and $\pi^* \in \Pi$ be any policy. Then, the choice $\lambda = 3\sqrt{3}\big(C_n(\Pi, M) + \log(30/\delta)\big)^{1/2}$ implies with probability at least $1 - \delta$ the following upper-bound on the excess risk*

$$L(\hat{\pi}^{CRM}) - L(\pi^*) \leq \sqrt{\frac{32V_M(\pi^*)\big(C_n(\Pi, M) + \log\frac{30}{\delta}\big)}{n}} + \frac{22S\big(C_n(\Pi, M) + \log\frac{30}{\delta}\big)}{n-1} + h_M(\pi^*)\,,$$

*where $V_M^2(\pi^*)$ and $h_M(\pi^*)$ are the variance and bias of the is the clipped estimator of $\pi^*$ and respectively defined in (26) and (29).*

*Proof.* We consider the notations of the proof of Proposition 4.1. Fix $\pi^* \in \Pi$. Applying, Theorem 15 of Maurer & Pontil (2009)[5] to the function set $\mathcal{F}_{\Pi,M}$ defined in (22), we get w.p. $1 - \delta$

$$\mathbb{E}_{(x,a,y)\sim\mathcal{P}_{\pi_0}}\big[f_{\hat{\pi}^{CRM}}(x, a, y)\big] - \mathbb{E}_{(x,a,y)\sim\mathcal{P}_{\pi_0}}\big[f_{\pi^*}(x, a, y)\big]$$

$$\leq \sqrt{\frac{32\mathrm{Var}_{(x,a,y)\sim\mathcal{P}_{\pi_0}}\big[f_{\pi^*}(x, a, y)\big]\big(C_n(\Pi, M) + \log\frac{30}{\delta}\big)}{n}} + \frac{22\big(C_n(\Pi, M) + \log\frac{30}{\delta}\big)}{n-1}\,.$$

Using the definition of $f_\pi(x, a, y)$ (22), we have

$$\mathbb{E}_{(x,a,y)\sim\mathcal{P}_{\pi_0}}\big[f_\pi(x, a, y)\big] = 1 + \frac{L^M(\pi)}{S} \text{ and } \mathrm{Var}_{(x,a,y)\sim\mathcal{P}_{\pi_0}}\big[f_\pi(x, a, y)\big] = \frac{V_M^2(\pi)}{S^2}\,,$$

where

$$V_M^2(\pi) = \mathrm{Var}_{(x,a,y)\sim\mathcal{P}_{\pi_0}}\left(y\zeta\left(\frac{\pi(a|x)}{\pi_0(a|x)}, M\right)\right)\,. \tag{26}$$

Substituting into the previous bound, this entails

$$L^M(\hat{\pi}^{CRM}) - L^M(\pi^*) \leq \sqrt{\frac{32V_M(\pi^*)\big(C_n(\Pi, M) + \log\frac{30}{\delta}\big)}{n}} + \frac{22S\big(C_n(\Pi, M) + \log\frac{30}{\delta}\big)}{n-1}\,. \tag{27}$$

To conclude the proof, it only remains to replace the clipped risk $L^M$ with the true risk $L$. On the one hand, since the costs $y$ take values into $[-1, 0]$, we have $y\zeta\left(\pi^*(a|x)/\pi_0(a|x), M\right) \geq y\pi(a|x)/\pi_0(a|x)$, which yields

$$L(\hat{\pi}^{CRM}) \leq L^M(\hat{\pi}^{CRM})\,. \tag{28}$$

On the other-hand, by defining the bias

$$h_M(\pi^*) = \mathbb{E}_{(x,a,y)\sim\mathcal{P}_{\pi_0}}\left[y\zeta\left(M, \frac{\pi^*(a|x)}{\pi_0(a|x)}\right) - y\frac{\pi^*(a|x)}{\pi_0(a|x)}\right] \tag{29}$$

we also have $-L(\pi^*) - h_M \leq -L^M(\pi^*)$, which together with (27) and (28) finally concludes the proof

$$L(\hat{\pi}^{CRM}) - L(\pi^*) \leq \sqrt{\frac{32V_M(\pi^*)\big(C_n(\Pi, M) + \log\frac{30}{\delta}\big)}{n}} + \frac{22S\big(C_n(\Pi, M) + \log\frac{30}{\delta}\big)}{n-1} + h_M(\pi^*)\,.$$

$\square$

We can now use the latter to prove Proposition 5.1 that is restated below.

---

[5]Note that in their notation, $\log\mathcal{M}_n(\pi)$ equals $C_n(\Pi, M) + \log(10)$, $\mathbf{X}$ is the dataset $\{(x_i, a_i, y_i)\}_{1\leq i\leq n}$ where $(x_i, a_i, y_i) \overset{i.i.d.}{\sim} \mathcal{P}_{\pi_0}$, and $P(\cdot, \mu)$ is the expectation with respect to one test sample $\mathbb{E}_{(x,a,y)\sim\mathcal{P}_{\pi_0}}[\cdot]$.

**Proposition 5.1.** *Consider the notations and assumptions of Proposition 4.1. Let $\hat{\pi}^{CRM}$ be the solution of the CRM problem in Eq. (15) and $\pi^* \in \Pi$ be any policy. Then, with well chosen parameters $\lambda$, denoting the variance $\sigma_*^2 = \mathrm{Var}_{\pi_0}\big[\pi^*(a|x)/\pi_0(a|x)\big]$, with probability at least $1 - \delta$:*

$$L(\hat{\pi}^{CRM}) - L(\pi^*) \lesssim \sqrt{\frac{(1+\sigma_*^2)\log(W+e)\big(C_n(\Pi,M) + \log\frac{1}{\delta}\big)}{n}} + \frac{\log(W+e)(C_n(\Pi,M) + \log\frac{1}{\delta})}{n} \, ,$$

*where $\lesssim$ hides universal multiplicative constants. In particular, assuming also that $\pi_0(x|a)^{-1}$ are uniformly bounded, the complexity of the class $\Pi_{CLP}$ described in Section 3.2, applied with a bounded kernel and $\Theta = \big\{\beta \in \mathbb{R}^m, \ s.t \ \|\beta\| \leq C\big\} \times \{\sigma\}$, is of order*

$$C_n(\Pi_{CLP}, M) \leq O(m\log n) \, ,$$

*where $m$ is the size of the Nyström dictionary and $O(\cdot)$ hides multiplicative constants independent of $n$ and $m$ (see (38)).*

*Proof.* We first consider a general policy class $\Pi$ and some $\pi^* \in \Pi$. In this proof, to ease the notation, we write $\mathbb{E}_{\pi_0}[\cdot]$, $\mathrm{Var}_{\pi_0}[\cdot]$, and $\mathbb{P}_{\pi_0}(\cdot)$ to respectively refer to $\mathbb{E}_{(x,a,y)\sim\mathcal{P}_{\pi_0}}[\cdot]$, $\mathrm{Var}_{(x,a,y)\sim\mathcal{P}_{\pi_0}}[\cdot]$, and $\mathbb{P}_{(x,a,y)\sim\mathcal{P}_{\pi_0}}(\cdot)$.

We consider the notation of the proof of Proposition E.1 and start from its risk upper-bound

$$L(\hat{\pi}^{CRM}) - L(\pi^*) \leq \sqrt{\frac{32V_M(\pi^*)\big(C_n(\Pi,M) + \log\frac{30}{\delta}\big)}{n}} + \frac{22S\big(C_n(\Pi,M) + \log\frac{30}{\delta}\big)}{n-1} + h_M(\pi^*) \, , \qquad (30)$$

where we recall, for any threshold $M$, the definitions of the bias and the variance of the clipped estimator of $\pi^*$,

$$h_M(\pi^*) = \mathbb{E}_{\pi_0}\left[y\left(\zeta\Big(\frac{\pi^*(a|x)}{\pi_0(a|x)}, M\Big) - \frac{\pi^*(a|x)}{\pi_0(a|x)}\right)\right] \text{ and } V_M^2(\pi^*) = \mathrm{Var}_{\pi_0}\left[y\zeta\Big(\frac{\pi^*(a|x)}{\pi_0(a|x)}, M\Big)\right].$$

**Step 1:** *For any threshold $M$, upper-bound of the variance $V_M(\pi^*)$ and the bias $h_M(\pi^*)$.*
By assumption, the (unclipped) variance of $\pi^*/\pi_0$ is bounded and we write

$$\sigma_*^2 = \mathrm{Var}_{\pi_0}\left[\frac{\pi^*(a|x)}{\pi_0(a|x)}\right] = \mathbb{E}_{(x,a,y)\sim\mathcal{P}_{\pi_0}}\left[\left(\frac{\pi^*(a|x)}{\pi_0(a|x)} - 1\right)^2\right].$$

First, we bound the clipped variance as

$$V_M^2(\pi^*) = \mathrm{Var}_{\pi_0}\left[y\zeta\Big(\frac{\pi^*(a|x)}{\pi_0(a|x)}, M\Big)\right] = \mathbb{E}_{\pi_0}\left[y^2\zeta\Big(\frac{\pi^*(a|x)}{\pi_0(a|x)}, M\Big)^2\right] - \mathbb{E}_{\pi_0}\left[y\zeta\Big(\frac{\pi^*(a|x)}{\pi_0(a|x)}, M\Big)\right]^2$$

$$\leq \mathbb{E}_{\pi_0}\left[\left(\frac{\pi^*(a|x)}{\pi_0(a|x)}\right)^2\right] - L^M(\pi^*)^2 = \sigma_*^2 + 1 - L^M(\pi_*)^2 \leq \sigma_*^2 + 1 \, . \quad (31)$$

Then, by writing $X = \pi^*(a|x)/\pi_0(a|x)$, the bias may be upper-bounded as

$$
\begin{aligned}
h_M(\pi^*) &\leq \mathbb{E}_{\pi_0}\big[X - \zeta(X, M)\big] \\
&\leq \mathbb{E}_{\pi_0}\big[(X - M)\mathbb{1}\{X > M\}\big] \\
&\leq \int_0^\infty \mathbb{P}_{\pi_0}\Big((X-M)\mathbb{1}\{X>M\} > t\Big)dt \\
&\leq \int_0^\infty \mathbb{P}_{\pi_0}\big(X - M > t\big)dt = \int_0^\infty \mathbb{P}_{\pi_0}\Big((X-1)^2 > (t+M-1)^2\Big)dt \\
&\leq \int_0^\infty \frac{\mathbb{E}_{\pi_0}\big[(X-1)^2\big]}{(t+M-1)^2}dt = \frac{\mathbb{E}_{\pi_0}\big[(X-1)^2\big]}{M-1} = \frac{\sigma_*^2}{M-1} \, .
\end{aligned}
\qquad (32)
$$

Furthermore, if $W \leq M$ then $S = \zeta(W, M) = W \leq M$, else, using $\alpha(M) = M/\log(\alpha(M)) \leq \max\{M, e\} \leq M + e$,

$$S = \zeta(W, M) = \alpha(M)\log\big(W + \alpha(M) - M\big) \leq (M + e)\log(W + e). \tag{33}$$

Therefore, substituting (31), (32), and (33) into (30), yields the following upper-bound on the excess risk

$$L(\hat{\pi}^{CRM}) - L(\pi^*)$$
$$\leq \sqrt{\frac{32(1 + \sigma_*^2)\big(C_n(\Pi, M) + \log\frac{30}{\delta}\big)}{n}} + \frac{22(M + e)\log(W + e)\big(C_n(\Pi, M) + \log\frac{30}{\delta}\big)}{n - 1} + \frac{\sigma_*^2}{M - 1}.$$

We now choose $M$ such that

$$\frac{22(M - 1)\log(W + e)\big(C_n(\Pi, M) + \log\frac{30}{\delta}\big)}{n - 1} = \frac{\sigma_*^2}{M - 1} \tag{34}$$

which is possible since the left term grows from 0 to infinity and the right term decreases from infinity to 0 for $M > 1$. Therefore, from the last two terms we eventually have

$$L(\hat{\pi}^{CRM}) - L(\pi^*) \lesssim \sqrt{\frac{(1 + \sigma_*^2)\log(W + e)\big(C_n(\Pi, M) + \log\frac{1}{\delta}\big)}{n}} + \frac{\log(W + e)(C_n(\Pi, M) + \log\frac{1}{\delta})}{n}, \tag{35}$$

where $\lesssim$ hides universal multiplicative constants. This concludes the first part of the proof.

**Step 2:** *Evaluating the policy class complexity $C_n\big(\Pi_{CLP}, M\big)$.*
In this part, we provide a bound on the metric entropy $C_n(\Pi_{\mathrm{CLP}}, M) = \log\mathcal{H}_\infty(1/n, \mathcal{F}_{\Pi_{\mathrm{CLP}}}, 2n)$. We recall that $\mathcal{F}_{\Pi_{\mathrm{CLP}}}$ is defined in (22) and $\Pi_{\mathrm{CLP}}$ is described in Section 3.2. More precisely, let $\mathcal{Z} \subseteq \mathcal{A}$ be a Nyström dictionary of size $m \geq 1$ and $\gamma > 0$. Since we use Gaussian distributions, we have

$$\Pi_{\mathrm{CLP}} = \big\{\pi_\beta \text{ s.t. for any } x \in \mathcal{X}, \ \pi_\beta(\cdot|x) = \mathcal{N}(\mu_\beta^{\mathrm{CLP}}(x), \sigma^2), \text{ with } \beta \in \Theta_\beta\big\},$$

where

$$\Theta_\beta = \big\{\beta \in \mathbb{R}^m, \text{ s.t } \|\beta\| \leq C\big\}$$

where

$$\mu_\beta^{\mathrm{CLP}}(x) = \sum_{a \in \mathcal{Z}} \frac{\exp(-\gamma\eta_\beta(x, a))}{\sum_{a' \in \mathcal{Z}} \exp(-\gamma\eta_\beta(x, a'))} \quad \text{and} \quad \eta_\beta(x, a) = \langle\beta, \psi(x, a)\rangle,$$

for some embedding $\psi$ described in Section 3.2 which satisfies $\|\psi(x, a)\| \leq \upsilon$ for any $(x, a)$. Fix $x \in \mathcal{X}$. Let us show that $\beta \mapsto \mu_\beta^{\mathrm{CLP}}(x)$ is Lipschitz. Denote by $Z_\beta(x) = \sum_{a \in \mathcal{Z}} \exp(-\gamma\eta_\beta(x, a))$ the normalization factor. We consider the gradient of $\mu_\beta^{\mathrm{CLP}}(x)$ with regards to $\beta$

$$\frac{\partial\mu_\beta^{\mathrm{CLP}}}{\partial\beta}(x) = \sum_{a \in \mathcal{Z}} a\bigg(\frac{\psi(x, a)\exp\big(\langle\beta, \psi(x, a)\rangle\big)}{Z_\beta(x)}$$
$$- \frac{\exp\big(\langle\beta, \psi(x, a)\rangle\big)\sum_{a \in \mathcal{Z}}\psi(x, a)\exp\big(\langle\beta, \psi(x, a)\rangle\big)}{Z_\beta(x)^2}\bigg).$$

Taking the norm, and upper-bounding $\|\psi(x, a)\| \leq \upsilon$ and $\|a\| \leq \alpha_{\mathcal{Z}}$, this yields

$$\bigg\|\frac{\partial\mu_\beta^{\mathrm{CLP}}}{\partial\beta}(x)\bigg\| \leq 2\upsilon\alpha_{\mathcal{Z}}.$$

Therefore, $\beta \mapsto \mu_\beta^{\mathrm{CLP}}(x)$ is $2\upsilon\alpha_{\mathcal{Z}}$-Lipschitz, which implies that

$$\beta \mapsto \pi_\beta(a|x) = \frac{1}{\sigma\sqrt{2\pi}}\exp\bigg(-\frac{1}{2}\bigg(\frac{a - \mu_\beta^{\mathrm{CLP}}(x)}{\sigma}\bigg)^2\bigg)$$

are also Lipschitz with parameter

$$\sqrt{\frac{2}{e\pi}}\frac{\upsilon\alpha_{\mathcal{Z}}}{\sigma^2}\,. \tag{36}$$

We recall that the metric entropy $C_{n,\delta}(\Pi_{\mathrm{CLP}}) = \log\mathcal{H}_\infty(1/n, \mathcal{F}_{\Pi_{\mathrm{CLP}}}, 2n)$ is applied to the function class

$$\mathcal{F}_{\Pi_{\mathrm{CLP}}} = \left\{ f_\beta : (x, a, y) \mapsto 1 + \frac{y}{S}\zeta\left(\frac{\pi_\beta(a|x)}{\pi_0(a|x)}, M\right) \quad \text{for some } \pi_\beta \in \Pi_{\mathrm{CLP}} \right\}.$$

By assumption, the inverse of the logging policy weights are bounded $\pi_0(a|x)^{-1} \le M_0$ for any $(x, a) \in \mathcal{X} \times \mathcal{A}$ (as in Kallus & Zhou (2018)). Therefore, together with (36), for any $(x, a, y) \in \mathcal{X} \times \mathcal{A} \times \mathcal{Y}$, the function $\beta \mapsto f_\beta(x, a, y)$ is Lipschitz with parameter

$$\sqrt{\frac{2}{e\pi}}\frac{\upsilon\alpha_{\mathcal{Z}}M_0}{S\sigma^2}\,. \tag{37}$$

Let $\varepsilon > 0$. Because there exists an $\varepsilon$-covering of the ball $\{\beta \in \mathbb{R}^d : \|\beta\| \le C\}$ of size $(C/\varepsilon)^d$, together with (37), the latter provides a covering of $\mathcal{F}_{\Pi_{\mathrm{CLP}}}$ in sup-norm with parameter

$$\varepsilon\sqrt{\frac{2}{e\pi}}\frac{\upsilon\alpha_{\mathcal{Z}}M_0}{S\sigma^2}\,.$$

Equalizing this with $n^{-1}$ and taking the log of the size of the covering entails

$$C_n(\Pi_{\mathrm{CLP}}, M) \le d\log\left(\sqrt{\frac{2}{e\pi}}\frac{CM_0\upsilon\alpha_{\mathcal{Z}}n}{S\sigma^2}\right).$$

Now, we recall that $d$ is the dimension of the embedding $\psi$, which we model as

$$\psi(x, a) = \psi_{\mathcal{X}}(x) \otimes \psi_{\mathcal{A}}(a)$$

where $\psi_{\mathcal{A}}(a) \in \mathbb{R}^m$ is the embedding obtained by using the Nyström dictionary of size $m$ on the action space and $\psi_{\mathcal{X}}(x) \in \mathbb{R}^{d_{\mathcal{X}}}$ is the embedding of the context space $\mathcal{X} \subseteq \mathbb{R}^{d_x}$. Typically $d_{\mathcal{X}} = d_x^2 + d_x + 1$ or $d_{\mathcal{X}} = d_x$ respectively with the polynomial and linear maps considered in practice. Thus, $d = md_{\mathcal{X}}$. Substituting the latter into the complexity upper-bound and using $1 \le S$ and $M$, we finally get

$$C_n(\Pi_{\mathrm{CLP}}, M) \le md_{\mathcal{X}}\log\left(\sqrt{\frac{2}{e\pi}}\frac{CM_0\upsilon\alpha_{\mathcal{Z}}n}{\sigma^2}\right), \tag{38}$$

where we recall that $d_{\mathcal{X}}$ is the dimension of the contextual feature map, $C$ a bound on the parameter norm $\beta$, $M_0$ a bound on $\pi_0(a|x)^{-1}$, $\upsilon^2$ a bound on the kernel, $\sigma^2$ the variance of the policies, and $\alpha_{\mathcal{Z}}$ a bound on the action norms $\|a\|$. $\qquad\square$

### E.2 Discussion: on the Rate Obtained for Deterministic Classes

Consider the deterministic CLP class that assigns any input $x$ to the action $\mu_\beta^{\mathrm{CLP}}(x)$ defined in (7). Although the paper focuses on stochastic policies, this appendix provides an excess-risk upper-bound with respect to this deterministic class.

The latter corresponds to the choice $\sigma = 0$ in the CLP policy set defined in Section 3.2 and therefore, Proposition 5.1 cannot be applied directly. Fix some $\sigma^2 > 0$ to be optimized later. For any $\beta \in \mathbb{R}^m$, we denote by $L(\mu_\beta^{\mathrm{CLP}})$ the risk associated with the deterministic policy $a = \mu_\beta^{\mathrm{CLP}}(x)$ and define $\pi_\beta^{\mathrm{CLP}}(\cdot|x) \sim \mathcal{N}(\mu_\beta^{\mathrm{CLP}}(x), \sigma^2)$. Then, let $\hat{\pi}^{\mathrm{CRM}}$ be the counterfactual estimator obtained by Proposition 5.1 on the class $\Pi_{CLP} = \{\pi_\beta^{\mathrm{CLP}}(\cdot|x)\}$, with probability $1 - \delta$

$$\hat{L}(\hat{\pi}^{CRM}) - L(\mu_\beta^{\mathrm{CLP}}) \le \hat{L}(\hat{\pi}^{CRM}) - L(\pi_\beta^{\mathrm{CLP}}) + L(\pi_\beta^{\mathrm{CLP}}) - L(\mu_\beta^{\mathrm{CLP}})$$

$$\le L(\hat{\pi}^{CRM}) - L(\pi_\beta^{\mathrm{CLP}}) + L_0\sigma\sqrt{2\log\frac{2}{\delta}},$$

where we assumed that the risk is $L_0$-Lipschtitz and used that $P\big(|X| < \sigma\sqrt{2\log(2/\delta)}\big) \le \delta$ for $X \sim \mathcal{N}(0, \sigma^2)$. From Proposition 5.1, this yields, with probability $1 - 2\delta$

$$\hat{L}(\hat{\pi}^{CRM}) - L(\mu_\beta^{\mathrm{CLP}}) \lesssim \sqrt{\frac{(1 + \sigma_*^2)\log(W + e)\big(C_n(\Pi_{\mathrm{CLP}}, M) + \log\frac{1}{\delta}\big)}{n}} + L_0\sigma\sqrt{2\log\frac{2}{\delta}}\,.$$

Now, note that $C_n(\Pi, M)$ and $\log(W)$ only yield logarithmic dependence on $\sigma^2$ and $n$ and will thus not impact the rate of convergence. The variance $\sigma^*$ has a stronger dependence on $\sigma^2$ but can be upper-bounded as follows

$$\sigma_*^2 = \mathrm{Var}_{\pi_0}\left[\frac{\pi_\beta^{\mathrm{CLP}}(a|x)}{\pi_0(a|x)}\right] = \int \left(\frac{\pi_\beta^{\mathrm{CLP}}(a|x) - \pi_0(a|x)}{\pi_0(a|x)}\right)^2 \pi_0(a|x)da$$

$$\le \int \frac{\pi_\beta^{\mathrm{CLP}}(a|x)^2}{\pi_0(a|x)}da \le \frac{1}{M_0}\int \pi_\beta^{\mathrm{CLP}}(a|x)^2 da = \frac{1}{2\sigma M_0\sqrt{\pi}}\,,$$

where the last equality is because $\pi_\beta^{\mathrm{CLP}}(\cdot|x)$ is a Gaussian distribution with variance $\sigma^2$. Therefore, keeping only the dependence on $\sigma$ and $n$ and neglecting log-factors, we get the high-probability upper-bound

$$\hat{L}(\hat{\pi}^{CRM}) - L(\mu_\beta^{\mathrm{CLP}}) \le \tilde{O}\Big(\frac{1}{\sqrt{\sigma n}} + \sigma\Big).$$

The choice $\sigma = n^{-1/3}$ entails a rate of order $\mathcal{O}(n^{-1/3})$.

## F   Details on the Experiment Setup and Reproducibility

In this section we give additional details on synthetic and semi-synthetic datasets, we provide details on the evaluation methodology and information for experiment reproducibility.

### F.1   Synthetic and Semi-Synthetic setups

**Synthetic setups**   As many real-world applications feature a reward function that increases first with the action, then plateaus and finally drops, we have chosen a piecewise linear function as shown in Figure 17 that mimics reward buckets over the CoCoA dataset presented in Section 6.

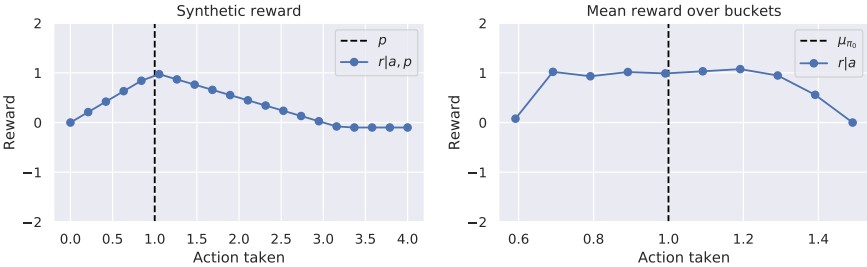

Figure 17: Synthetic reward engineering. The synthetic reward (left) is inspired from real-dataset reward buckets (right).

We provide an illustration of the three synthetic datasets in Figure 18.

**Semi-synthetic medical setup**   The semi-synthetic cost inputs prescriptions from medical experts to obtain $y(a, x) = \max(|a - t^*| - 0.1t^*, 0)$, so as to mimic the expert prediction. The logging policy $\pi_0$ samples actions $a \sim \pi_0$ contextually to a patient's body mass index (BMI) score $Z_{BMI} = \frac{x_{\mathrm{BMI}} - \mu_{\mathrm{BMI}}}{\sigma_{\mathrm{BMI}}}$ and can be analytically written with i.i.d noise $e \sim \mathcal{N}(0, 1)$, moments of the therapeutic dose distribution $\mu_T^*, \sigma_T^*$ such that $a = \mu_T^* + \sigma_T^*\sqrt{\theta}Z_{BMI} + \sigma_T^*\sqrt{1 - \theta}\varepsilon$ ($\theta = 0.5$ in the setup of Kallus & Zhou (2018)). The logging probability density function thus is a continuous density of a standard normal distribution over the quantity $\frac{a - \mu_T^* + \sigma_T^*\sqrt{\theta}Z_{BMI}}{\sigma_T^*\sqrt{1 - \theta}}$.

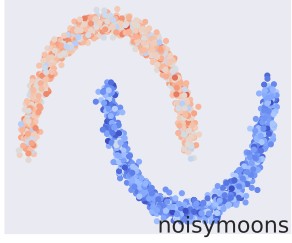 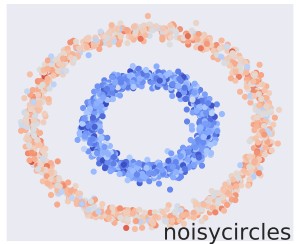 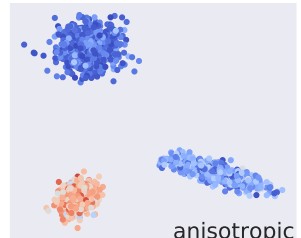

Figure 18: Contexts (points in $\mathbb{R}^2$), and potentials represented by a color map for the synthetic datasets. Learned policies should vary with the context to adapt to the underlying potentials.

## F.2 Reproducibility

We provide code for reproducibility and all experiments were run on a CPU cluster, each node consisting on 24 CPU cores (2 x Intel(R) Xeon(R) Gold 6146 CPU@ 3.20GHz), with 500GB of RAM.

**Policy parametrization.** In our experiments, we consider two forms of parametrizations: (i) a lognormal distribution with $\theta = (\theta_\mu, \sigma), \pi_{(\mu,\sigma)} = \log \mathcal{N}(m, s)$ with $s = \sqrt{\log(\sigma^2/\mu^2 + 1)}; m = \log(\mu) - s^2/2$, so that $\mathbb{E}_{a\sim\pi_{(\mu,\sigma)}}[a] = \mu$ and $\mathrm{Var}_{a\sim\pi_{(\mu,\sigma)}}[a] = \sigma^2$; (ii) a normal distribution $\pi_{(\mu,\sigma)} = \mathcal{N}(\mu, \sigma)$. In both cases, the mean $\mu$ may depend on the context (see Section 4), while the standard deviation $\sigma$ is a learned constant. We add a positivity constraint for $\sigma$ and add an entropy regularization term to the objective in order to encourage exploratory policies and avoid degenerate solutions.

**Models.** For parametrized distributions, we experimented both with normal and lognormal distributions on all datasets, and different baseline parameterizations including constant, linear and quadratic feature maps. We also performed some of our experiments on low-dimensional datasets with a stratified piece-wise contextual parameterization, which partitions the space by bucketizing each feature by taking $K$ (for e.g $K = 4$) quantiles, and taking the cross product of these partitions for each feature. However this baseline is not scalable for higher dimensional datasets such as the Warfarin dataset.

**Hyperparameters.** In Table 5 we show the hyperparameters considered to run the experiments to reproduce all the results. Note that the grid of hyperparameters is larger for synthetic data. For our experiments involving anchor points, we validated the number of anchor points and kernel bandwidths similarly to other hyperparameters.

| | Synthetic | Warfarin | CoCoA |
|---|---|---|---|
| Variance reg. $\lambda$ | $\{0., 0.001, 0.01, 0.1, 1, 10, 100\}$ | $\{0.0001 0.001 0.01 0.1\}$ | $\{0., 0.001, 0.1\}$ |
| Clipping $M$ | $\{1, 1.7, 2.8, 4.6, 7.7, 12.9, 21.5, 35.9, 59.9, 100.0\}$ | $\{1, 2.1, 4.5, 9.5, 20\}$ | $\{1, 2.1, 4.5, 9.5, 10, 20, 100\}$ |
| Prox. $\kappa$ | $\{0.001, 0.01, 0.1, 1\}$ | $\{0.001, 0.01, 0.1\}$ | $\{0.001, 0.01, 0.1\}$ |
| Reg. param. $C$ | $\{0.00001, 0.0001, 0.001, 0.01, 0.1\}$ | $\{0.00001, 0.0001, 0.001, 0.01, 0.1\}$ | $\{0.00001, 0.0001, 0.001, 0.01, 0.1\}$ |
| Number of anchor points | $\{2, 3, 5, 7, 10\}$ | $\{5, 7, 10, 12, 15, 20\}$ | $\{2, 3, 5\}$ |
| Softmax $\gamma$ | $\{1, 10, 100\}$ | $\{1, 5, 10\}$ | $\{0.1, 0.5, 1, 5\}$ |

Table 5: Table of hyperparameters for the Synthetic and CoCoA datasets

# G Additional Results and Additional Evaluation Metrics

In this section we provided additional results on both contextual modeling and optimization driven approaches of CRM.

## G.1 Continuous vs Discrete strategies in Continuous-Action Space

We provide in Figure 19 additional plots for the continuous vs discrete strategies for the synthetic setups described in Section 7.2.1.

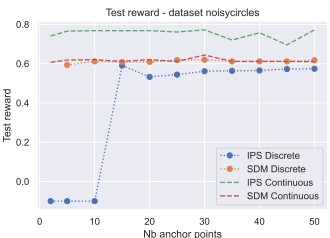 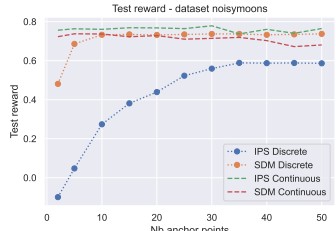 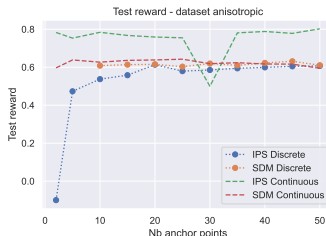

Figure 19: Continuous vs discrete. Test rewards for CLP and (stochastic) direct method (DM) with Nyström parameterization, versus a discrete approach, with varying numbers of anchor points. We add a minimal amount of noise to the deterministic DM in order to pass the $n_{\text{eff}}$ validation criterion.

## G.2 Soft clipping improves or competes with other importance weighting transformation strategies

**Soft-clipping improves or competes with other importance weighting transformation strategies.** We also experiment on the synthetic datasets comparing our soft clipping approach with other methods which focus is to improve upon the classic clipping strategy for the same optimization purposes. Notably, we consider the power-mean correction of importance sampling (PMCIS) (Metelli et al., 2021) method which we adapt to our continuous modeling strategy to enable fair comparison. Moreover, we also added the SWITCH (Wang et al., 2017) as well as the CAB (Su et al., 2019) methods. However, both methods use a direct method term in their estimation, which is difficult to adapt for stochastic policies with continuous actions, as explained in our discussions on doubly robust estimators (see Appendix G.4). Therefore, we considered discretized strategies and compared them with soft clipped estimator applied to discretized policies. For the discretized strategies, we have used the same anchoring strategies as described before, namely using empirical quantiles of the logged actions (for 1D actions), and have optimized the number of anchor points along with the other hyperparameters using the offline evaluation protocol. We see overall in Table 6 that our soft-clipping strategy provides satisfactory performance or improves upon all weight transforming strategies on the synthetic datasets.

|  | Noisymoons | Noisycircles | Anisotropic |
|---|---|---|---|
| Logging policy $\pi_0$ | 0.5301 | 0.5301 | 0.4533 |
| SWITCH (discrete) | $0.5786 \pm 0.0025$ | $0.5520 \pm 0.0026$ | $0.5741 \pm 0.0024$ |
| CAB (discrete) | $0.5761 \pm 0.0024$ | $0.5534 \pm 0.0025$ | $0.5705 \pm 0.0021$ |
| scIPS (discrete) | $\mathbf{0.5888} \pm 0.0022$ | $\mathbf{0.5637} \pm 0.0024$ | $\mathbf{0.5941} \pm 0.0019$ |
| PMCIS (CLP) | $0.7244 \pm 0.0005$ | $0.7189 \pm 0.0004$ | $\mathbf{0.7739} \pm 0.0008$ |
| scIPS (CLP) | $\mathbf{0.7674} \pm 0.0008$ | $\mathbf{0.7805} \pm 0.0004$ | $0.7703 \pm 0.0002$ |

Table 6: Comparison of importance weight transformations on the synthetic datasets, for discretized strategies and for continuous action policies.

## G.3 Optimization Driven Approaches of CRM

In this part we provide additional results on optimization driven approaches of CRM for the Noisycircles, Anisotropic, Warfarin and CoCoAdatasets.

Both Noisycircles and Anisotropic datasets in Figure 20 show the improvements in test reward and in training objective of our optimization-driven strategies, namely the soft-clipping estimator and the use of the proximal point algorithm. Overall we see that for most configurations, the proximal point method better optimizes the objective function and provides better test performances, while the soft-clipping estimator performs better than its hard-clipping variant, which may be attributed to the better optimization properties. For semi-synthetic Warfarin and real-world CoCoA datasets in Figure 20 we also show the improvements in test reward and in training objective of our optimization-driven strategies. More particularly we demonstrate the

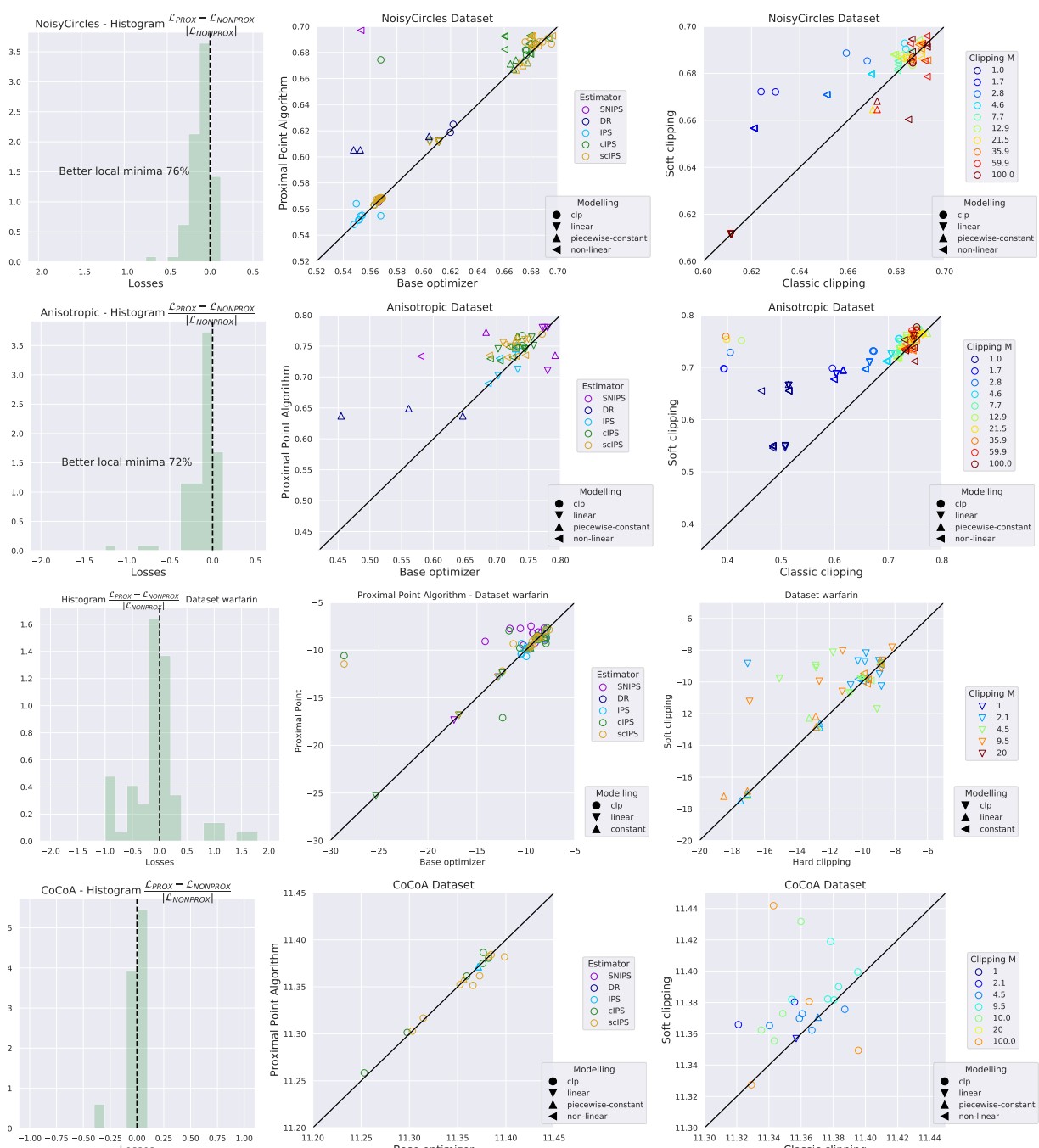

Figure 20: Optimization-driven approaches (NoisyCircles, Anisotropic, Warfarin and CoCoA datasets). Relative improvements in the training objective from using the proximal point method (left), comparison of test rewards for proximal point vs the simpler gradient-based method (center), and for soft- vs hard-clipping (right).

effectiveness of proximal point methods on the Warfarin dataset where most proximal configurations perform better than the base algorithm. Moreover, soft-clipping strategies perform better than its hard-clipping variant on real-world dataset with outliers and noises, which demonstrate the effectiveness of this smooth estimator for real-world setups.

**On further optimization perspectives and offline model selection** As mentioned in Section 4, our use of the proximal point algorithm differs from approaches that enhance policies to stay close to the logging policies which modify the objective function as in (Schulman et al., 2017) in reinforcement learning. Another avenue for future work would be to investigate distributionnally robust methods that do such modifications of the objective function or add constraints on the distribution being optimized. The policy thereof obtained would thus be closer to the logging policy in the CRM context. Moreover, as we showed in Section 6.2 with importance sampling estimates and diagnostics, the offline decision becomes less statistically significant as the evaluated policy is far from the logging policy. Investigating how the distributionally robust optimization would yield better CRM solutions with regards to the offline evaluation protocol would make an interesting future direction of research.

### G.4 Doubly Robust Estimators

In this section we detail the discussion on doubly robust estimators and the difficulties that exist to obtain a suitable estimator. In policy based methods for discrete actions, the DR estimator takes the form

$$\hat{L}_{\text{DR}}(\pi) = \frac{1}{n}\sum_{i=1}^{n} \frac{\pi(a_i|x_i)}{\pi_{0,i}}\left(y_i - \hat{\eta}(x_i, a_i)\right) + \frac{1}{n}\sum_{i=1}^{n}\sum_{a\in\mathcal{A}}\pi(a|x_i)\hat{\eta}(x_i, a)$$

Usually, the DR estimator should only improve on the vanilla IPS estimator thanks to the lower variance induced by the outcome model $\hat{\eta}$. However, in a continuous-action setting with stochastic policies, the second term becomes $\mathbb{E}_{x\sim\mathcal{P}_{\mathcal{X}}, a\sim\pi(\cdot|x)}\left[\hat{\eta}(x, a)\right]$, which is intractable to optimize in closed form since it involves integrating over actions according to $\pi(\cdot|x)$. Thus, handling this term requires approximations (as described hereafter), which may overall lead to poorer performance compared to an IPW estimator that sidesteps the need for such a term.

The difficulty for stochastic policies with continuous actions is to derive an estimator of the term $\mathbb{E}_{x\sim\mathcal{P}_{\mathcal{X}}, a\sim\pi(\cdot|x)}\left[\hat{\eta}(x, a)\right]$. Unlike stochastic policies with discrete actions which allow to use a discrete summation over the action set, we would need here to compute here an estimator of the form $\frac{1}{n}\sum_{i=1}^{n}\int_{a\in\mathcal{A}}\pi(a|x_i)\hat{\eta}(x_i, a)$. We note that in the case of deterministic policy $\pi$ learning this direct method term would easily boil down to $\frac{1}{n}\sum_{i=1}^{n}\hat{\eta}(x_i, \pi(x_i))$, and the DR estimator would be built with smoothing strategies for the IPW term as in (Kallus & Zhou, 2018).

In our experiments for stochastic policies with one dimensional actions $\mathcal{A}\subset\mathbb{R}$, we tried to approximate the direct method term $\frac{1}{n}\sum_{i=1}^{n}\int_{a\in\mathcal{A}}\pi(a|x_i)\hat{\eta}(x_i, a)$ with a finite sum of CDFs differences over the $m$ anchor points $a_1, \ldots a_m$ by computing :

$$\frac{1}{n}\sum_{i=1}^{n}\sum_{j=1}^{m}\int_{a=a_j}^{a_{j+1}}\pi(a|x_i)\hat{\eta}(x_i, a_j)$$

We present a table below of some of the experiments we ran on the synthetic datasets we proposed, along with an evaluation of the baselines that exist in the litterature for discrete actions. We see overall that our model improves indeed upon the logging policy, but does not compare to the performances of the scIPS and SNIPS estimators.

Demirer et al. (2019) provide a doubly robust (DR) estimator on continuous action using a semi-parametric model of the policy value function. Their policy learning is performed in two stages (i) estimate a doubly robust parameter $\theta^{DR}(x, a, r)$ in the semi-parametric model of the value function $\mathbb{E}[y|a, x] = V(a, x) = \langle\theta_*(x), \phi(a, x)\rangle$ and (ii) learn a policy in the empirical Monte Carlo estimate of the policy value by solving

$$\min_{\pi\in\Pi}\left\{\hat{V}^{DR}(\pi) := \frac{1}{n}\sum_{i=1}^{n}\langle\hat{\theta}^{DR}(x_i, a_i, r_i), \phi(\pi(x_i), x_i)\rangle\right\}.$$

|  | Noisymoons | Noisycircles | Anisotropic |
|---|---|---|---|
| Logging policy $\pi_0$ | 0.5301 | 0.5301 | 0.4533 |
| Doubly Robust (discrete) | $0.5756 \pm 0.0022$ | $0.5500 \pm 0.0024$ | $0.5593 \pm 0.0026$ |
| SWITCH (discrete) | $0.5786 \pm 0.0025$ | $0.5520 \pm 0.0026$ | $0.5741 \pm 0.0024$ |
| CAB-DR (discrete) | $0.5683 \pm 0.0023$ | $0.5326 \pm 0.0025$ | $0.5361 \pm 0.0028$ |
| Doubly Robust (ours) | $\mathbf{0.6115} \pm 0.0001$ | $\mathbf{0.6113} \pm 0.0002$ | $\mathbf{0.5977} \pm 0.0001$ |

Table 7: Comparison of doubly robust estimators, discretized strategies and our model which approximates the direct method term.

The doubly robust estimation is performed with respect to the first parameter learned in (i) for the value function, while we follow the CRM setting Swaminathan & Joachims (2015a) and directly derive estimators of the policy value (risk) itself, which would correspond to the phase (ii). Instead, to derive an estimate a DR estimator of such policy values, we tried extending the standard DR approach for discrete actions from Dudik et al. (2011) to continuous actions by using our anchors points, but these worked poorly in practice.

