# OpenReview forum: "Counterfactual Learning of Stochastic Policies with Continuous Actions"
_TMLR — Accepted by TMLR_

### Review · Reviewer_p3aZ · 2024-06-13

**Summary Of Contributions:**

I will start by briefly summarizing my understanding of the counterfactual risk minimization learning problem setting, since I am not well-acquainted with the related literature. Essentially, we have $n$ IID triplets $(x\_{1},a\_{1},y\_{1}),\\ldots,(x\_{n},a\_{n},y\_{n})$ where each $x_{i}$ (a feature vector of some kind) denotes some "context" for the decision maker to choose an action, which in the case of $x\_{i}$, is $a\_{i}$. Under this context and action, a loss $y\_{i}$ is incurred. We (i.e., the learning algorithm) has access to these triplets, but does *not* know anything about the underlying "policy" upon which each action $a\_{i}$ was selected (a potentially noisy mapping from $x\_{i}$. To the best of my understanding, the interpretation of this $n$-sized dataset is information about "what actually happened under some unknown policy," and what we want to know is "*what would happen* if we used a certain policy $\\pi \\in \\Pi$?" (without running the expensive/dangerous/questionable experiments that would be needed to actually generate new triplets based on a candidate $\\pi$). This "what-if" nuance aligns with the term "counter-factual" used in the paper. The ultimate goal is then to optimize the expected loss to be incurred under a given policy, but again with the critical constraint that we cannot re-run the (action-dependent) loss-generating process; this constraint is the critical difference between the concept of "losses" in standard empirical risk minimization, where losses can be re-computed at training time for any choice of policy.

With my above understanding of the basic problem of interest as context, my understanding of the authors' contribution in this submission is as follows. The main deficiency in the existing literature used as a springboard to motivate this work is a lack of effective methods when the action space (from which the policy samples) is *continuous*. In this direction, they propose the use of distributions (conditional on "context" $x$) which are characterized by mean and variance (see equation (6)). From my perspective, their key modelling decision is to fix the variance and set the mean as a function of $x$ by taking a weighted sum of "anchor points" (a finite subset of actions) whose weights are determined by loss predictions made by an ancillary "loss predictor" $\\eta\_{\\beta}(x,a)$ which can be trained based on the available context-action-loss triplets. The authors also touch on this predictor in some detail, considering a kernel-based joint embedding of actions and contexts.

Somewhat orthogonal to the above-mentioned model proposal is the authors' consideration of how to actually select an "optimal" policy from their newly designed model. There are two facets to this: the objective function to use, and the optimization scheme to use. Regarding the objective function, the starting point is the classic IPS method (see their equation (4)), but this can have serious volatility issues when the probability in the denominator is close to zero. A natural solution is clipping these importance weights (which leads to bounded losses), but clipping can reduce the efficacy of gradient-based optimizers since the loss is constant beyond the clipping threshold (i.e., zero slope). The authors thus consider a soft clipping approach; their penalty values become unbounded, but gradient information is more meaningful. They complement this proposal with learning theoretical analysis; they give one-sided tail bounds for their empirical objective (yielding upper bounds on the ideal objective; see Prop 4.1), as well as risk bounds for minimizers of their empirical objective function (modified to include an empirical variance term; see Prop 5.1). These results mostly follow in a mechanical way from well-established arguments (especially since they just assume their way out of unbounded losses), but I think makes a nice complement to the exposition of their approach.

Finally, they also have put significant effort into empirical tests (sections 6 and 7). One side of this is a new dataset based on real-world bidding data, plus a protocol for evaluation when using such data. The other side is evaluating the efficacy of their approach using both controlled simulations and the aforementioned real data evaluation procedure. The authors conclude that their proposed combination of model/optimization methods tends to be superior to more naive traditional approaches.

**Audience:**

Yes

**Broader Impact Concerns:**

No relevant concerns.

**Claims And Evidence:**

Yes

**Requested Changes:**

Overall, I don't have any critical requests, just a handful of points I tripped up on. I hope that at least one of the other reviewers is a bit more knowledgeable than I am regarding standard experimental design in the counterfactual learning setting.

- Page 2, "findings" list, point 3: the authors state that *"Real-world counterfactual risk minimization requires a standard offline evaluation protocol,"* but this is not really a finding, just a statement of opinion. The authors have a proposed protocol in section 6; can they say anything about its validity/transparency/utility relative to other plausible protocols? I think this kind of content is what readers would expect in a "findings" list.

- Page 3: a "logged" dataset and a "logging" policy; are these standard terms? I had no idea what was meant until I finished reading the following page and inferred the overall problem setting.

- Page 4: *"The expected risk of a policy..."* This should be expected loss or just risk, since later on the authors say *"...with small risk."*

- Page 4, after equation (2), *"When using counterfactual estimators for $\\widehat{L}$..."*. As a new reader to this domain, the term "counterfactual estimator" was totally impenetrable to me; even now, I'm not really sure how to digest this term. Estimators of the expected loss when we cannot obtain new losses as a function of action? (and only have access to the aforementioned "logged" triplets?)

- After equation (4), the first sentence is quite poor. The statement *"has large variance"* comes with no qualification, and the statement *"is subject to various overfitting phenomena"* is too vague to have much meaning. The following sentence I also found troublesome; what do you mean by *"negative"* feedback values? Simply being below zero, or being *far* below zero?

- Section 3.2: the dimension number $d\_{x}$ for $x$; in my opinion it is bad practice to use one symbol $x$ for effectively two distinct purposes, one as a random variable, and one as a symbol to indicate what the dimension number is for. Simply $d$ would be better if it hasn't been used elsewhere.

- I felt the anchor set used in (7) and the fixed variance in (6) are pretty critical, non-trivial design decisions, but they are done with minimal comment. A bit more discussion might be quite beneficial in my opinion.

- Top of Page 8, *"...we recall the empirical variance with scIPS that is...": the meaning of "scIPS" is not defined anywhere in the preceding text. As far as I can tell, it just appears as a subscript on the left-hand side of (12).

- Page 8, *"...compared to a linear dependence for IPS"*: please provide a clear citation to the relevant literature where such a result can be found.

- Remark 4.1: the term "costs" is used, but I assume this refers to losses? It would be best to keep formal terms consistent.

- Section 6: is it "CoCoA dataset" or "CoCoAdataset" (without a space)? The sub-section heading for 6.1 has a space, but immediately afterward the space is gone.

- Page 11: *"propensity over-fitting"* --> *"propensity __for__ over-fitting"*.

- Section 8: For future research, the authors mention that they would *"like to discuss the doubly robust estimator"*? Is discussing something really future research? The last sentence of the conclusions section feels like a throwaway sentence, and deserves to be a bit more well thought through.

**Strengths And Weaknesses:**

__Strengths:__

The main strength of this paper is that the authors have found a practically useful approach for a difficult learning problem, and this is backed up by encouraging results on a wide variety of setups including real-world data that they have curated themselves. I am not at all familiar with the related literature, so I have low confidence in my evaluation of the validity of their experimental protocols, but overall the paper is quite well written with lucid exposition, and I felt that they did a good job of placing their proposed methods (sections 3 through 5) in the proper context given existing work. There is nothing particularly surprising about the learning theoretical results, nor do they capture optimization error in their analysis, but I think the results given definitely contribute to the overall exposition. Overall, their claims appear quite transparent, and look to be sufficiently well-rooted in the empirical evidence provided.


__Weaknesses:__

For better or for worse, this is in my opinion applied machine learning research, and their main contributions are *empirical* in nature. This need not be a weakness to be sure, since they have found a combination of techniques which demonstrably work, which will assuredly be valuable to practitioners trying to tackle (continuous) counterfactual learning problems, but the *insights* obtained here are perhaps somewhat limited. This is just because the positive empirical performance was obtained as the result of aggregating a handful of different techniques (estimated loss-weighted average over "anchor" grid, soft-clipped importance weights, proximal point process), making it hard to pin down exactly what matters (in terms of modelling/algorithm design decisions) and when.

---

> ### Author Response · Authors · 2024-12-03
> **Empirical nature and practical insights**
>
> We thank the reviewer for their valuable feedback and for recognizing the practical value of our work, which was initially motivated by a real-world problem. We acknowledge that our contribution has a significant empirical component, and we believe this is crucial for bridging theory and practice in the literature, providing actionable techniques for practitioners.
>
> We also recognize the challenge of isolating the contributions of individual components in our proposed method. To address this, we conducted detailed analyses: We examined the impact of our clipping strategy independently across all modeling approaches (our CLP, as well as linear and kernel modeling; see Figure 5). We studied the influence of the PPA estimator alone across all modeling approaches (see Figure 6). Finally, we compared the CLP modeling to other approaches under the best optimization settings (see Table 3).
>
> This led us to derive key insights in terms of modelling (joint context-action embeddings, like CLP, are more informative for learning), optimizing (differentiable clipping and regularized learning provide better solutions) and evaluating policies for real-world applications (like the CoCoA dataset) and for practitioners who want to validate their learning beyond simulations.
>
> We have addressed the requested changes below and have updated the manuscript accordingly:
> - Page 2, evaluation protocol: To the best of our knowledge, our evaluation protocol is novel and does not directly compare to existing protocols in this context. We emphasize the importance of having a robust evaluation procedure and have highlighted a significant gap in the literature, where some methods have been developed and validated exclusively on simulated data.
> - Page 3 “logging” and “logged” : the terminology indeed is derived from Swaminathan et al, 2012: we have added a footnote to clarify what it refers to with regards to the importance sampling literature (Owen, 2016).
> - Page 4, “expected risk”: we have corrected the expression into “risk”.
> - Page 4, after equation (2), counterfactual estimators: we have added a footnote. The "counterfactual" term comes from the causal inference literature and refers in this policy learning context to all estimators using the inverse propensity scoring (IPS) method (Horvitz & Thompson,;1952) which corresponds to importance sampling in our continuous action setting
> - After equation (4) we have reformulated our point in explaining two different kind of overfitting in the counterfactual logged bandit feedback learning setting : depending on whether the losses are positive or negative, the learned policy can either overfit or avoid training data to artificially minimize the loss in propensity overfitting.
> - dimension d_x, we have modified and standardized the notation to d
> - anchor points (7) and variance (6):  we have incorporated further details on the variance design at the beginning of Section 3.2 and on discussion on anchor points that was at the end of Section 3.2. We also added a sentence on their empirical selection in Section 7.2.1.
> top page 8, meaning of scIPS: we have clarified the text just before Eq. 13 to provide an inline definition of scIPS.
> - page 8, we have added a citation of (Swaminathan et al, 2012) where the linear dependence on the clipping parameter is given.
> remark 4.1: we changed costs into losses
> - section 6.1 CoCoA dataset, it is with a space, we have corrected the two typos.
> - page 11: propensity overfitting: the term propensity in the literature refers to the probability density function (and the probability in discrete actions spaces) of the sampled action given the context. We added a brief description of what propensity overfitting is and also added a reference to Section 3.1 where it now has a clearer description.
> - Section 8 discussion: we have clarified what was intended in further investigating the doubly robust estimator for future research.

---

> > ### Comment · Reviewer_p3aZ · 2024-12-06
> > **Re: Empirical nature and practical insights**
> >
> > I thank the authors for their response, and acknowledge the changes made.

---

### Review · Reviewer_HdeN · 2024-06-27

**Summary Of Contributions:**

The submission discusses the problem of counterfactual learning of stochastic policies with continuous actions. The main contribution includes (1) a new modeling for counterfactual learning w.r.t. continuous actions, based on a joint kernel embedding of contexts and actions; (2) a novel dffirentiable estimator for counterfactual risk, based on soft-clipping technique on IPS, with generalization bound, and a corresponding optimizer based on proximal point methods; (3) a new released large-scale real-world dataset, for continous actions, and corresponding offline evaluation procedure for model selection in CRM problem.

**Audience:**

Yes

**Claims And Evidence:**

Yes

**Requested Changes:**

I do not see much necessary changes and recommend acceptence as it is.

**Strengths And Weaknesses:**

Strengths:
1. The modeling approach for continous actions is novel and inspiring for CRM problem.
2. Concrete theoretical proof for the new proposed estimator.
3. Well-organization for the paper presenation.

Weaknesses:
1. The idea of soft-clipping IPS, its proofs, and the evaluation protocol for logged data are kind of standard.

---

> ### Author Response · Authors · 2024-12-03
> **Soft clipping and evaluation protocol novelty**
>
> We thank the reviewer for their positive review and for highlighting the clarity of our presentation, as well as the novelty of our modeling. While the proof of our analysis indeed follows techniques from (Maurer and Pontil, 2009), the proposed soft-clipping estimator remains a novel approach which differs from existing importance weighting transformation strategies (Wang et al, 2017), (Su et al, 2019) and (Metelli et al, 2021) which motivation were not on the differentiability of the estimator. As for the offline evaluation protocol, to the best of the authors' knowledge, the literature on the offline logged bandit feedback problem does not yet establish a standard evaluation procedure, which would be necessary for real-world applications. We believe that the development and formalization of such protocols are crucial for advancing the field. We kindly ask the reviewer to suggest any papers that propose potential evaluation protocols in this context.

---

### Review · Reviewer_UPsJ · 2024-11-19

**Summary Of Contributions:**

The paper builds ideas on top of the problem of counterfactual learning of stochastic policies from logged bandit feedback. In particular, the manuscript considers the scenario where actions are continuous, which can be considered more or less harder. Two main technical contributions are added to the previous SOTA methods: i) the addition of a joint kernel embedding of context and actions, plus a Nystrom approximation to provide a finite-dimensional framework to the problem, and ii) a differentiable soft-clipping strategy such that optimization in CRM is stable and feasible.

**Audience:**

Yes

**Claims And Evidence:**

Yes

**Requested Changes:**

I believe the paper in its current version has a severe problem with clarity. This is certainly my main concern, as I feel that in this way it should not be considered for acceptance, as it would be really difficult for readers to dig into it and its proposed ideas.

First, the structure chosen for the thread of explanations is really difficult to follow and to understand what is going on. At times, one has to re-read sections several times to not get lost in the hyper-connected sentences about new equations, terms, parameters and threshold hyperparameters. In that sense, I could even feel that the text loses several times the thread and the goal of what it wants to communicate. What I want to say also, is that the way of dividing the manuscript into titled paragraphs is not helping at all. For example, I can clearly see that the “summary of the CLP policy” on pp.7 is added as a way of clarifying after one realizes that pp.5-6 are not a well-structured flow of continuously changing methods, and ideas, and approximations and techniques. These two pages do really need re-writing, otherwise it’s impossible to get what is going on.

This way of presenting scientific contributions makes also difficult to understand why some decisions are taken. A good example of parts where this is a bit better indeed is the part of modeling the loss predictor, with the kernel embedding and why the Nystrom approximation with its anchor points is needed. But even here, it should be clearly explained why the joint kernel embeddings make sense (I get it, but from my own thinking, not because it is explicitly mentioned in the paper — and I do think manuscripts like these should be self-contained and make basically things easier for the reader).

Last but no least, I add some comments and details that make me be a bit concerned, and perhaps are good indicators of where new updates can be made:

- After Eq. (4) — I understand that the empirical estimator might have a large variance, but why it is prone/subject to various overfitting phenomena? isn’t this counter-intuitive?

- End of pp.5: what does it mean to avoid the optimization over actions? Could it be added something else to this point?

- End of pp.6:  Anchor points, maybe a little larger discussion on discretization would be nice as it touched one of the main topics of the paper (continuous actions over discrete ones)

- Summary in pp.7: I do believe it is redundant: and basically adding it is a result of the lack of clarity

- Section 4.1 on soft-clipping IPS and why makes it non-differentiable. I think this part is a strength, but it is oddly written in my opinion. Can be improved quite a lot.

- I don’t really know if Eq. (11) is a great contribution or just an added method to make it differentiable in the larger regime. Could authors help here to provide more insights?

- I find it weird that generalization bounds are introduced just after the analysis on optimization in that way —> ok I see that it is pretending to justify its use as a good optimization objective for minimizing the expected risk, but it creates a super large jump in the text.

**Strengths And Weaknesses:**

The manuscript is in a positive direction, and kind of includes all key elements that a work of this sort should have. Some points that I consider worth mentioning:

- The presentation of the problem is decent, even if it could be a bit more clear and straightforward, I could get what the authors wanted to do.

- The review and consideration of other SOTA methods is fine, at least, I do not feel something relevant is missing. However, I have some concerns about the similarities wrt Swaminathan and Joachims  (ICML 2015), as I perceive that it is the key reference work for the manuscript. To me, evaluating the degree of difference and contributions wrt that work is difficult, and at times, the lack of clarity and concise comments on the new contributions makes it a bit harder.

- Technically, I think derivations in the main manuscript are well-developed, as I didn’t find any clear mistakes or errors. The decisions taken on the side of the kernel embedding and the Nystrom approximation make sense to me.

(I will include my points of concern and thoughts on weaknesses in the following section [requested changes])

---

> ### Author Response · Authors · 2024-12-03
> **Clarity concerns, rewriting**
>
> We thank the reviewer for their valuable feedback. We have revised the manuscript accordingly (with changes highlighted in blue), enhancing the presentation and clarifying the flow of ideas. Specifically, we addressed several issues in the presentation. Motivated by a real-world application, we needed to tackle three independent aspects of the problem—modeling, estimation and optimization, and evaluation—which resulted in three sections that initially appeared disconnected. In this revision, we have clarified the connections between these sections, particularly in the introduction and Section 3.1. Additionally, we addressed the lack of clarity and motivation for the CLP model. We believe the revised presentation in Section 3.2 provides a clearer explanation of the model and its significance. The final point, which may be seen as questionable, is the placement of the generalization bounds after the optimization section. This choice was motivated by the need to provide a theoretical justification for the optimization objective used to minimize the risk.
>
> Our work builds upon the framework proposed by Swaminathan and Joachims (ICML 2015), which is designed for discrete actions. However, adapting this framework to real-world datasets with continuous actions yields several challenges across multiple dimensions. These issues, which we address in our approach, are outlined at the end of the introduction and in Section 3.1 of the revised manuscript.
>
> The motivation for adopting a joint kernel embedding framework is twofold: (i) While naive discretization or bucketization strategies often fail to deliver satisfactory results, kernel embeddings provide a natural framework for modeling smooth functions (here, of actions and contexts); (ii) Jointly modeling actions and contexts allows us to approximate the optimal deterministic policy described in Eq. (5). In contrast, existing approaches, such as those proposed by Chen et al. (2016) and Kallus & Zhou (2018), use kernels only on contexts. This limitation reduces the model expressivity in many applications, as shown in our experiments (e.g., see Table 2). These points are now more clearly explained in Section 3.2. Additionally, based on the reviewer’s suggestion, we have removed the titled paragraphs in the presentation of the CLP model and replaced them with clearer subsections.
>
> We have also read the other concerns on clarity and addressed the following points:
> - After Eq. 4, overfitting phenomena: we have specified two kinds of overfitting (loss overfitting and propensity overfitting) that may occur in the CRM setting with the classical IPS estimator.
> - End of pp5 : optimization over actions: we have reformulated the paragraph and specified how the soft-argmin operator approximately minimizes the policy risk with the finite anchor points.
> - End of pp6: we have extended the discussion on the discretization and the number of anchor points and have clarified the reference to - Appendix G.1 where an empirical comparison between continuous and discrete strategies is provided.
> - Summary in pp7: We have removed the summary at the end of the section and replaced it with an algorithm box that consolidates and illustrates all the steps for constructing our continuous action modeling in between the explanations.
> - Section 4.1 soft-clipping is differentiable unlike the hard-clipping strategy as it uses a differentiable log operator. We have added a figure on the soft clipping to illustrate the effect of this soft-clipping operator.
> - Eq. (11) represents a novel approach that is different from existing importance weighting transformation strategies, such as those proposed by Wang et al. (2017), Su et al. (2019), and Metelli et al. (2021). Unlike these methods, which do not prioritize the differentiability of the estimator, our approach explicitly addresses this aspect. We have added this clarification and updated the paper to reference Appendix G.2 for further details.
> - Generalization of the error bound after the analysis: the generalization error bound indeed serves as illustrating how minimizing the upper bound of the risk would generate a policy which does minimize the true risk.

---

### Decision · Action_Editor_C13e · 2025-02-14

**Recommendation:** Accept as is

**Comment:**

This paper presents a novel approach to counterfactual learning with continuous actions, addressing a critical challenge in offline policy learning. The key contributions include a joint kernel embedding framework for modeling counterfactual policies, a differentiable soft-clipping estimator to stabilize optimization, and a theoretical analysis establishing generalization bounds. Additionally, the paper introduces a new large-scale real-world dataset, which enhances the empirical validation of the proposed method and provides a valuable resource for future research. The combination of these elements results in a comprehensive approach to counterfactual risk minimization in the continuous action setting.

The motivation for accepting this work stems from its strong technical foundation, thorough empirical evaluation, and practical relevance. The proposed approach effectively tackles an important problem in the field, improving both theoretical understanding and practical applicability. Reviewers found the manuscript to be well-supported by rigorous analysis and experiments, demonstrating clear benefits over existing methods. While initial concerns regarding clarity and organization were raised, the revised version successfully addressed these issues, leading to unanimous agreement among the reviewers that the paper is now in an acceptable state for publication. Given the significance of the contributions and the improvements made in response to feedback, we are pleased to accept your paper for publication.

**Audience:**

Yes, the findings of this paper would be of interest to at least some individuals in TMLR's audience. The paper addresses a fundamental challenge in counterfactual learning with continuous actions, a topic relevant to researchers in machine learning, reinforcement learning, and causal inference. The proposed method introduces novel techniques for improving policy learning from logged data, which is highly applicable in domains such as healthcare, online advertising, and recommendation systems—areas where counterfactual reasoning plays a crucial role.

**Claims And Evidence:**

Yes, the claims made in the submission are supported by accurate, convincing, and clear evidence. The paper provides a strong theoretical foundation, including well-developed mathematical derivations and generalization bounds that validate the proposed method. The empirical evaluation is thorough, incorporating both synthetic experiments and a newly introduced large-scale real-world dataset, which strengthens the credibility of the results.